# Forcing Pulsations by Means of a Siren for Gas Turbine Applications

**Fabrice Giuliani [1,*]**, **Markus Stütz [1]**, **Nina Paulitsch [1]** and **Lukas Andracher [2]**

[1] Combustion Bay One e.U., Advanced Combustion Management, 8010 Graz, Austria; Markus.Stuetz@CBOne.at (M.S.); Nina.Paulitsch@CBOne.at (N.P.)

[2] Institute of Aviation, Department of Engineering, University of Applied Sciences FH JOANNEUM GmbH, 8020 Graz, Austria; Lukas.Andracher@fh-joanneum.at

[*] Correspondence: Fabrice.Giuliani@CBOne.at; Tel.: +43-316-22-8980

**Abstract:** A siren is a robust fast-valve that generates effective flow pulsations and powerful noise levels under well-controlled conditions. It operates under the inlet flow conditions of a gas turbine combustor. Its principle is based on a sonic air jet periodically sheared by a cogged wheel rotating at a given speed. It is used as an alternative to loudspeakers in combustion laboratories when the use of these is made difficult by aggressive flow conditions, such as hot air under pressure, possibly containing impurities. It is also a serious candidate as an effective flow actuator to be deployed on power gas turbine fleets. The authors have gathered more than twenty years of knowledge on siren technology. This pulsator was originally developed for research on thermoacoustics. By scanning through a given frequency range, one detects the acoustic resonance of specific parts of the combustor assembly, or possibly triggers a combustion instability during a sensitivity analysis of a flame to small perturbations. In 2010, Giuliani et al. developed a novel siren model with the capacity to vary the amplitude of pulsation independently from the frequency. In this contribution, the physics, the metrics, and the resulting parameters of the pulsator are discussed. Technical solutions are unveiled about visiting large frequency ranges (currently 6 kHz) and achieving elevated pressure fluctuations (150 dB SPL proven, possibly up to 155 dB SPL) with a compact device. A multimodal excitation is available with this technology, one idea being to dissipate the acoustic energy on nearby peaks. The contribution ends with a summary of the applications performed so far and the perspective of an industrial application.

**Keywords:** combustion instabilities; forcing; siren; flow control; calibration

## 1. Introduction

This paper is about active control of combustion stability in gas turbines. In opposition to batch-cycle internal combustion engines, continuous flow machines, such as a gas turbine, are supposed to operate a steady flow conditions at the different stations of the Joule-Brayton cycle. The turbulent flow can be described by a stochastic bandwidth of eddies and boundary layers, in which sizes go from typical passages length down to micro-structures according to the Kolmogorov scales, following the flow. However, if some conditions are met, the flow can turn unsteady, resulting in strong pulsations at a given frequency. The combustor is prone to provide these conditions, and the problem of combustion instabilities has become more acute with low-emission burners operating in the lean domain.

The conditions to achieve a resonance in the flow are a resonator, a trigger, an energy source, and an arrangement favouring a coupling of all of these. Resonance is usually achieved with sudden section changes, or obstacles in the flow connected to cavities of a characteristic length or volume, so that specific frequencies are enhanced in bulk mode, or Helmholtz mode. The trigger can be the

turbulent flow itself, for instance, a vortex detachment of von Kármán street type at a section change or behind an obstacle. The trigger can also be the flow noise related to the rotor speed, the stator–rotor interaction, and the passage through specific parts that resonate when excited at their eigenfrequencies. Sound waves propagate from their point source in all directions, at the vector sum of the flow speed plus the sound speed [1]. In relationship with the turbulent flow pattern, the machine's noise is also spread over a given bandwidth. The energy term in the case of combustion instabilities is the unsteady part of the heat input. The latter depends on mixing time scales, thermochemical time scales, and specific flame lengths scales, also resulting in a given bandwidth of combustion turbulence. Taken separately, the energy content of each unsteadiness is very low compared to the flow's kinetic energy, overall pressure, or thermal energy. They are named small perturbation.

Combustion instabilities appear when all of these conditions happen to interact at some point. This involves an aero-thermo-acoustic coupling, which is one way to describe physically the combustion instability. Small perturbations can therefore trigger a large-amplitude fluctuation of each of these quantities, where the steady flow becomes pulsed in the combustor, where the local pressure fluctuation is such that vibrations and narrow-peak sounds are being generated, and the flame front deforms and moves periodically at the same pace possibly nearer to the walls. The sum of it all greatly impairs the performance of the machine. It also accelerates the hot core's fatigue in a dramatic manner. It raises dramatic safety concerns. This explains why modern gas turbine combustors are equipped with high-temperature resistant fast-pressure sensors and accelerometers so that the combustion stability is monitored in real time. Should an instability be detected at an early stage, the operation is tuned so that this instability disappears.

Combustion instabilities are the reason for the development of the apparatus discussed in this paper: a siren. In the early years of space exploration, the topic was combustion instabilities in rocket engines [2]. Then, it became critical in large power gas turbines operating at elevated operation ratio and under lean conditions, a pattern which is conducive to the presence of combustion instabilities [3,4]. The problem of combustion instability has been a major theme of research over the last half century, as well as a major barrier against the introduction of lean burn technologies in aeroengines to this day [5].

Numerous investigations on the experimental side, as well as in terms of models, were conducted to make progress on the understanding of combustion instabilities [6,7], as well as on technologies to avoid or damp these when they happen (passive and active combustion control [8,9]), or to benefit from this coupling (pulse combustion [10]). The present paper reviews different experimental techniques used to reproduce the flow patterns described before in a well-controlled manner. By opposition to self-resonating situations, the instability is hereby forced. The paper starts with a general overview and then focuses on one particular technology developed by authors over the two last decades. Different modes of operations are described. The blow-down forcing is the most effective one, where the siren is placed upstream from the pressurised test cell. As a comparison, an effective configuration of discharge forcing is shown: the siren is placed downstream from the test cell or derives a part of the pressurised air in it. The second configuration is a convenient strategy for the control of the acoustic boundaries in the combustor because it does not relies on a separate source of compressed air anymore, but it leaks periodically a part of the plenum air that can be derived further down the machine, e.g., for cooling purposes. In laboratories, the siren can be used as a calibrator for fast-pressure probes or accelerometers. Another application is to force flow pulsation and trigger, in a well-controlled manner, thermoacoustic instabilities on research flames. In the industry, the intention is to equip gas turbine combustors with an effective flow modulator for control purposes, which has the possibility to pass through some critical frequencies with a zero-gain. One application is the displacement of the acoustic energy from a resonant frequency to one of its sub-harmonics, where the selected sub-harmonic is not harmful. A more immediate application is to generate a powerful and calibrated sound and then check out the response and sensitivity drift of all mounted pressure sensors and accelerometers with time, without dismounting them, and possibly during operation. We show and explain why multimodal excitation makes sense. Suggestions are made in the end about the use of this equipment for sensor

calibration, instrumentation health monitoring and for active combustion control purposes in power gas turbines.

## 2. Solutions for Flow Modulation, Noise Emissions, or both Simultaneously

Pulse combustion applications, as well as the challenge of combustion instabilities, explain why devices for dynamic flow control were developed, with the intention to understand the physics involved in couplings of thermoacoustic nature and eventually control them. Pulse combustion [10,11] covers a range of rather rare applications exploiting the beneficial effects of thermoacoustic couplings. Combustion instabilities are the uncontrolled pendant of pulse combustion. They happen due to an inconvenient coupling of dynamic parameters under some operating conditions of the machine.

### 2.1. Passive and Active Control

Regarding commercial systems, the favoured solution is passive control, where the machine does not rely on an additional apparatus. It involves a refined hot core design with smoothly shaped transitions that avoids acute resonant frequencies (e.g., that could be triggered by the flow itself). For ad-hoc solutions, pacification elements, such as acoustic liners, Helmholtz resonators, or wall absorbers, can be used to avoid or damp critical resonances. However, as long as passive control is not satisfactory—and in case the system has already been deployed and suffers on-site from a combustion instability never seen before, as has happened several times in history, one must rely on an ad-hoc solution based on active control.

Active control is about controlled flow forcing. The forcing is a deliberate action performed on a continuous flow, with the intention to bring in a calibrated periodic compound. Acting at selected frequencies by means of an additional apparatus called pulsator can bring a system out of balance that would tend to be naturally stable. The frequency must match an eigenfrequency, and the level of perturbation must overcome a threshold to initiate the resonance. After that, the level of pulsation grows up to the limit cycle amplitude, where the forces of excitation acting on the growth rate are counterbalanced by damping effects [12].

Active control is considered where passive control measures are ineffective. In a laboratory, active control delivers well-controlled boundary conditions [6–9]. In the industry, ad-hoc active control systems were developed to solve grave combustion instability problems [3,4].

The difficulty of active control is the intrusion of a separate aggregate in the gas turbine machinery, the need for fail-safe strategies and the difficulty in transmission of the know-how from one device to the next. Keeping that in mind, the siren apparatus detailed in the following was developed with an all-rounder philosophy for laboratory and for power GT applications.

### 2.2. Active Control for Combustion in Continuous Flow Machines

The air inlet valve of the Argus-Schmidt pulse jet is one historical piece of technology designed to periodically let in a mass flow of fresh air into a quarter-wave thermoresonator called Schmidt-Argus tube [11]. This device was part of the V1-"buzz bomb" during World War II [13]. The valve consists in an array or flexible metal blades, comparable in shape and in vibration dynamics to clarinet mouthpieces, slightly bent and pressed two by two against each other. These are called Reed valves (see Figure 1). They were designed to open/close at pressure fluctuations up to 3% of the ambient pressure at about 43 Hz. Based on this cyclic air feed and on a pulse combustion process in a quarter-wave Schmidt tube, the pulse jet could develop from 2 to 3.3 kN of thrust. The periodic valve was therefore an ad-hoc design able to sustain the combustion pulsation process.

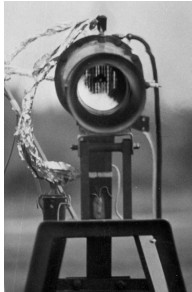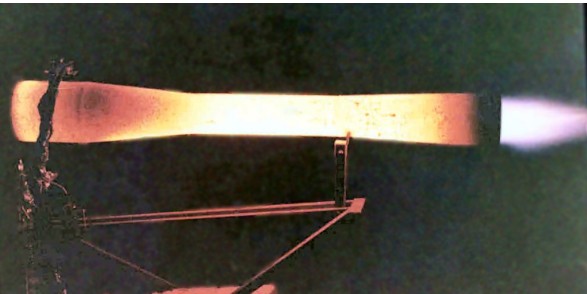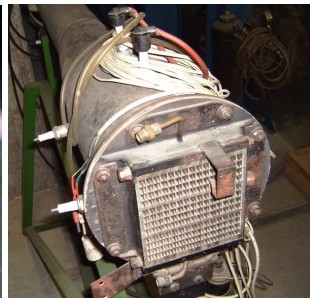

**Figure 1.** Pulse combustion application: the Schmidt-Argus pulse jet engine in action. Thrust tests led at Thalerhof airport by Zhuber and Pirker in the 1960s (archives TU Graz/TTM) with detail of the Reed air valve and ignition sparklers.

The low pressure ratio achieved was also the limiting factor in terms of performance, and this explains why this principle was not further developed for propulsion. Nevertheless, some important works by Reynst [14] on a better heat generation making the best possible use of thermoacoustics have led, for instance, to a few commercial applications of pulse combustion boilers; the unique selling proposition (USP) of these being a better thermal efficiency than conventional, a complete combustion and a satisfying trade-off regarding emissions. Modern applications show that the problems inherent to noise, vibrations, and limited operational bandwidth have been managed with some success over time [15].

Combustion instabilities are the uncontrolled pendant of pulse combustion. They happen due to an inconvenient match of parameters at some place of the machine's operational bandwidth. This explains why actuators were developed to reproduce these conditions, with the intention to understand the physics of the interaction, and eventually control them.

Many combustion laboratories perform successful flow modulation by means of a loudspeaker, or a set of loudspeaker [16–18]. For power systems, applications using loudspeakers are unrealistic because of the aggressive operating conditions (high pressure, high temperature), on the one side, and because of their limited acoustic power, on the other.

The first successful industrial combustion stability control systems were acting on the injection of fuel, such as Moog's D6xx series Direct-Drive Valves (DDV) used by Hermann et al. [3].

Previous works, such as author's about multiphase, show that the most promising strategy is an actuation performed on the transporting phase, namely the air [19]. Technologies others than loudspeakers have been tried: rotating valves by Choudhury et al. [20], confined pulse jet by Parikh [21] based on a similar principle, electro-pneumatic valves use by Yu et al. [22], or mechanical shakers with Burnel et al. [23]. In order to provide a satisfactory trade-off between operational range and effective pulsation in presence of aggressive flow conditions, the siren technology was chosen.

### 2.3. The Siren

The layout of a siren consists in a jet of air sheared periodically by a rotating disk by means of teeth or holes on this disk. Some models of siren were developed for operation under elevated conditions of flow pressure and temperature, specifically for large mass flow rates at the cost of low pressure drops (about 4%) [24,25]. In the following, we discuss a siren technology with a large pressure drop and a critical jet, setting precisely the amount of average and pulse flows based on the sole upstream conditions [26]. In comparison to the references mentioned previously, the passage of sheared air is unique and the cooling of the motor part is simpler. Since this siren technology is based on a sheared sonic jet, the resulting pulsation is depending on the generating conditions only. In other words, it acts independently from a possibly established instability situated downstream, which is convenient for control purposes.

### 2.4. Evolution of a Pulsator of Siren Type

#### 2.4.1. The ONERA Siren

A first model developed at ONERA in the frame of the studies [6,19,27] is an evolution of electro-pneumatic devices taking into consideration an hot air flow up to 500 K and elevated pressure operation. The motivation was to develop a compromise between a rotating valve (for a wide pulsation frequency range) and a shocked nozzle with varying critical surface for a high air flow rate. The device included a 20 teeth cogged wheel placed below a sonic hole and covering periodically its initial surface of 15%. The pulsation range could cover to 0–1 kHz at 15% pulsation amplitude. The wheel was entrained by a constant-current electromotor type ESCAP 460 SC, non-regulated mounted in direct drive with the cogged-wheel. The device is represented on Figure 2.

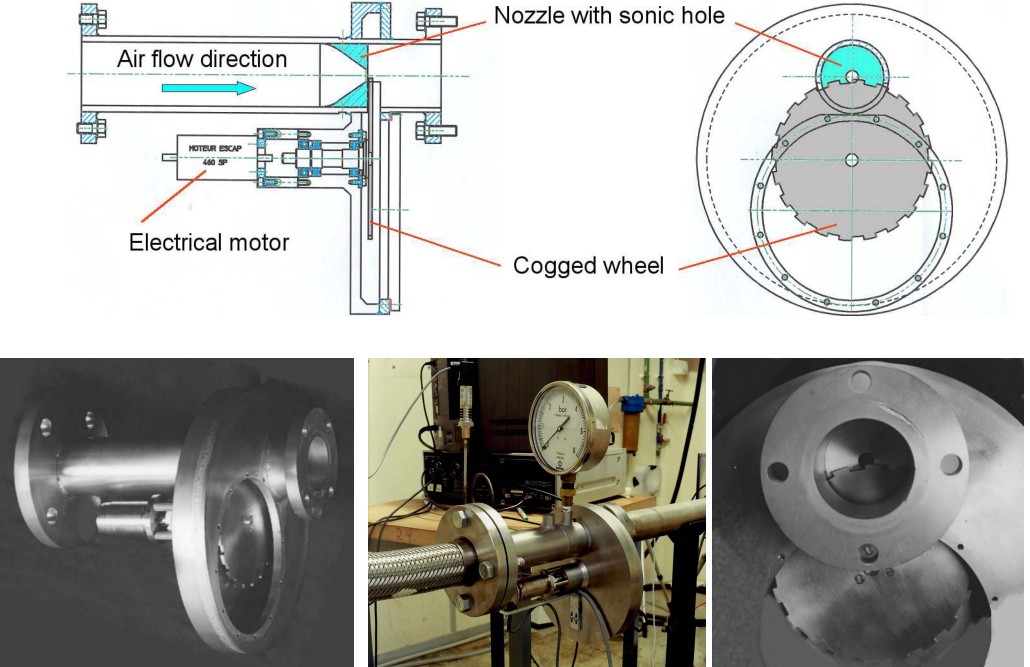

**Figure 2.** The ONERA siren [6,19,27]. (**Up**): Side and front views of the actuator. (**Down**): Views including open wheel casing.

The siren reports in real time about the pulsation status. The phase-defined feedback is provided by a secondary wheel placed on the same entrainment axis as the cogged wheel, as shown on Figure 3. This wheel possesses as many holes as there are teeth on the cogged wheel. An optoelectronic sensor placed close to its edge detects the passage of a hole. This provides the TTL phase reference signal (or triggering signal for the acquisition systems). This could drive, for instance, a phase-locked stroboscope, or any apparatus able to sample data at one precise moment of the pulsation.

The same device was adapted on all further siren generations. It operates well on the investigated 0–6 kHz range.

#### 2.4.2. TU Graz Siren

TU Graz performed from 2004 to 2010 a research programme on the density fluctuation in pulsed flames and used the first apparatus in publications [28–30]. The basic principle, as well as the feedback, repeats the ONERA siren features. Improvements were wished in terms of precision, this is why a servo motor was implemented. It was also observed that the fixed pulsation rate could be problematic at some frequencies where the flame would blow out because of the strength of the pulsation. The second-generation of the siren can be seen in Figure 4, where details of the different mechanisms are shown.

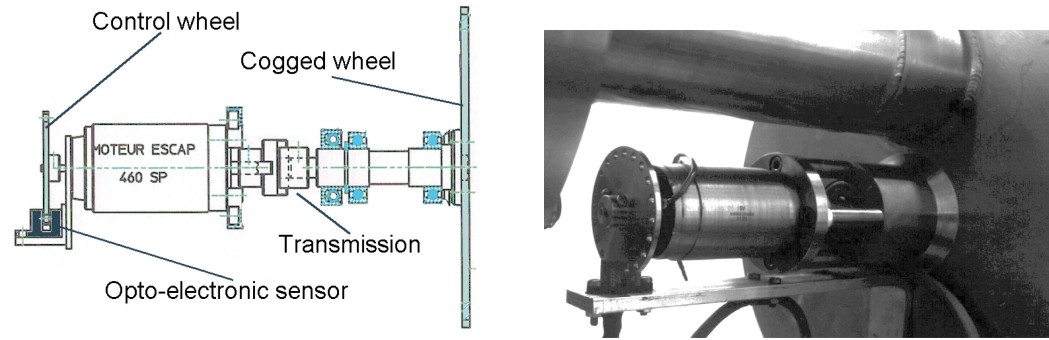

**Figure 3.** Optoelectronic sensor for real-time frequency and phase information on the air pulse state. (**Left**): Technical drawing. (**Right**): Close-up of the device mounted at the rear of the air pulsator carter.

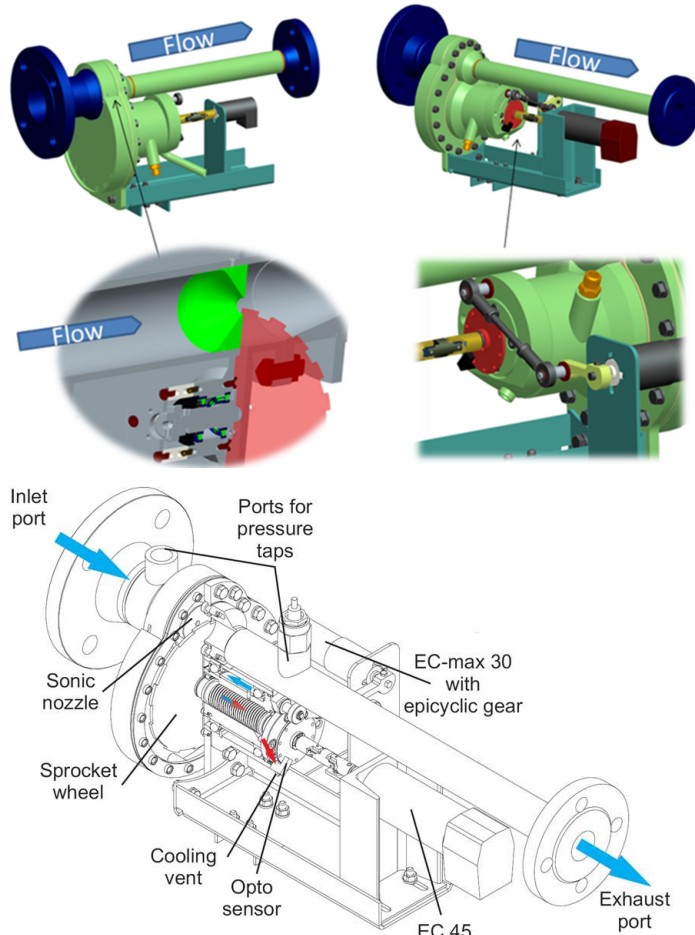

**Figure 4.** TU Graz siren, with details of the nozzle and wheel assembly, as well as the connecting rod, able to change the free critical surface based by moving the eccentric shaft of the cogged wheel.

Therefore, a system able to vary the amplitude was designed, based on the refined positioning of the cogged wheel relatively to the fix-mounted nozzle by means of an adjustable eccentric frame. The drive of the cogged wheel would be transmitted by a cardan to the wheel's shaft. The drive of the adjustable eccentric frame is driven by a connecting rod. With this device, one could go through critical frequencies at zero gain amplitude and therefore pass through these frequencies without damage, which is convenient. The principle is shown in Figure 5.

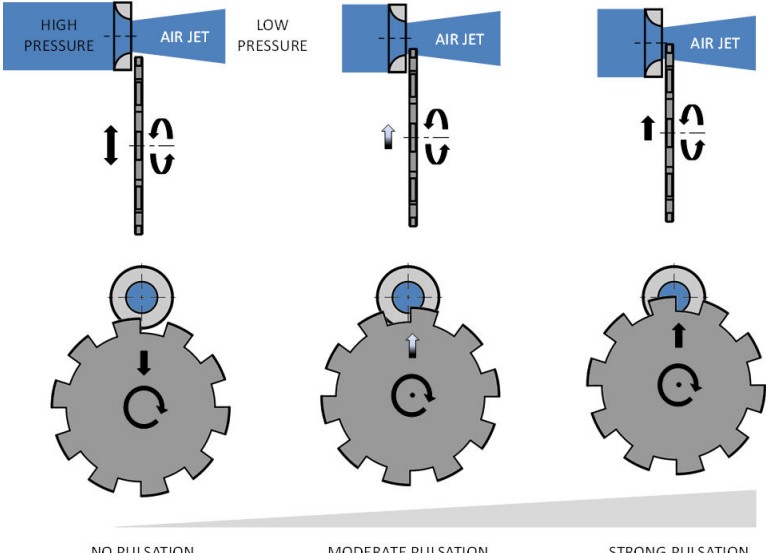

**Figure 5.** Principle of changing the amplitude of pulsation by moving the position of the cogged wheel's shaft relatively to the nozzle.

### 2.4.3. Combustion Bay One (CBOne)'s Siren

The TU Graz pulsator's description published in Reference [26] triggered the interest of other laboratories. Additional features were discussed, such as the possibility to "tune up the volume" in a refined way as one would do with a loudspeaker, as well as generate well-repeatable pulsation conditions if possible with help of a pre-programmed batch-file. Based on that demand, it was decided that a spin-off would be created to produce and provide this pulsator, and that was the reason for being of the company Combustion Bay One e.U. (founded 2012). The siren was therefore redesigned to gain in performance (higher flow rates, higher temperatures, higher frequency ranges), be practical in use (pre-programmed batch file to repeat sequences of tests, improved interface), and more flexible (trade-off nozzle and wheel as a function of the desired operation, and eased interchangeability of these). CBOne's siren model 3G (third generation) is shown in Figure 6. One detail of importance is the gear replacing the connection rod for more precision on the amplitude.

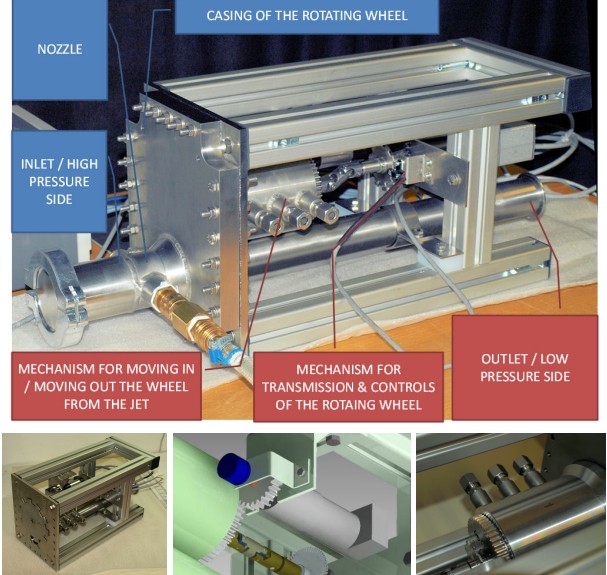

**Figure 6.** Combustion Bay One (CBOne)'s siren model 3G [31]: Main features (**top**) and details (**bottom**) showing the open casing, the amplitude gear, and the cooling ports.

## 3. Elements of Design

### 3.1. Mass Flow

The siren's concept described before is based on the discharge of a sonic jet. The mass flow of a sonic jet is dependent on the generating conditions ($P_o$, $T_o$) and on the nozzle neck section $S$ only. This area is called the critical section, and under these conditions the nozzle is choked (the Mach number is 1 in this section). This area is the quantity that is being periodically varied with the siren. From then on, the mass flow is proportional to the generating pressure $P_o$ under isothermal conditions. The geometry of the convergent and critical section is calculated according to Hugoniot's formulas. There is no divergent; the jet blows in the open of the wheel casing. The assumptions are that the head conditions ($P_o$, $T_o$) are constant. A critical section can also be used as a mass flow meter provided the nozzle is choked, and that its dimensions and the generating conditions are known. One relies on the perfect gases' law. The other assumptions are that the small distance separating the jet critical section from the plane of the wheel is negligible, the thickness of the wheel compared to the diameter of the jet is negligible, and the specific relaxation times (or response time) of a critical jet are much smaller that the shear periods, i.e., the flow response is instantaneous. The mass flow in the critical section is the following:

$$\dot{m} = \sqrt{\frac{\gamma}{r\,T_o}}\,P_o\left(\frac{2}{\gamma+1}\right)^{\frac{\gamma+1}{2(\gamma-1)}} S, \tag{1}$$

$$= \sqrt{\frac{\gamma}{r\,T_o}}\,P_o\left(\frac{2}{\gamma+1}\right)^{\frac{\gamma+1}{2(\gamma-1)}} \frac{\pi d^2}{4} \quad \text{for a round nozzle.} \tag{2}$$

When using air ($\gamma$ = 1.4), the simplified formula is:

$$\dot{m} = \frac{0.6847}{\sqrt{r\,T_o}}\,P_o\,S$$

$$= \frac{0.1712\,P_o\,\pi\,d^2}{\sqrt{r\,T_o}} \quad \text{for a round nozzle.}$$

### 3.2. Technical Solution to Achieve Elevated Frequencies

Solutions were developed to augment the pulsation performance while keeping the pulsator's dimensions similar. Over the last years, new requests including higher frequencies and higher pulsation levels have been formulated. The higher frequency is needed to visit up to the second harmonic of precessing vortex cores [18,32], or spinning tangential instabilities [33]. Therefore, we discuss effective pulsations in the range zero to six kHz. The second motivation is to achieve pulsation levels near 155 dB SPL for calibration purposes, and at higher ambient pressures and temperatures compared to standard conditions. If the chocked nozzle's surface is limited in size, the alternative is to augment the backpressure. The walls, casing, and seals can be adapted accordingly.

For a given maximum motor speed, elevated pulsation frequencies can be achieved by maximising the number of teeth on the cogged wheel. However, a limiting factor on this tooth number is related to the ratio between the average distance between two upward fronts on the wheel (or tooth period) and the nozzle width. This ratio must remain larger than 2; otherwise, a cut-off frequency is met. This is the reason why one considers slot-like nozzles as an arrangement to this limiting factor.

A new geometry tending towards a slot was developed as an alternative to the round critical nozzle of diameter $d$. A double-Laval convergent profile was drawn along the two axes of symmetry of the slot to ensure that the nozzle surface is choked at the upper and lower edges, as shown in Figure 7.

| Surface ratio of disc ø $d$ | 3/4 | 1/2 | 1/4 |
|---|---|---|---|
| Slot shape | | | |
| Width (% $d$) | 63.5% | 40.4% | 19.8% |
| Surface of circumscribing rectangle $d * h$ | 92.8% | 97.2% | 99.3% |

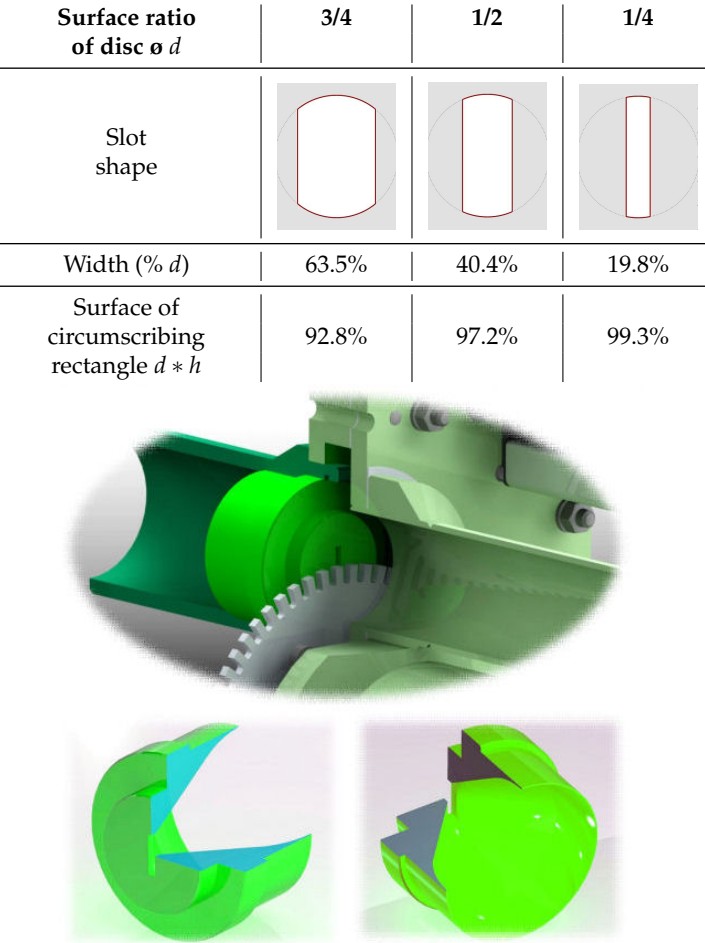

**Figure 7.** Novel nozzle profiles showing a slot-like outlet, for several round surface ratios (**top**), nozzle arrangement with wheel (**middle**), and nozzle cut views (**bottom**).

It is of importance to relate to a round surface to easily connect to the set of round nozzles. The resulting slot therefore has a surface corresponding to a given ratio of the reference disc area with diameter $d$. For a ratio $1/x$ of the reference round surface $\pi d^2/4$, the width $w$ of the slot is:

$$
\begin{aligned}
w &= d \sin \alpha \\
\text{where} \quad \alpha \quad &\text{is the solution of } \sin \alpha \cos \alpha + \alpha - \frac{\pi}{2x} = 0.
\end{aligned}
\tag{3}
$$

Figure 7 shows the nozzle surface for ratios 3/4, 1/2 and 1/4. Slots having a surface equal to or smaller than 1/4 of the reference disc area can be approximated by a rectangle of dimensions $d$ and $w$. The same figure shows an arrangement with a 60-teeth wheel able to achieve 5.4 kHz with the current model. A set of quarter-round nozzles was produced with success using additive manufacturing [34].

### 3.3. Multimodal Excitation

The incentive is to augment the number of excited frequencies. The intention is twofold. In this paper, it is shown that at least three distinct frequencies can be generated by the siren and accelerate the characterization of the resonance of the system, by gathering more resonances in one take while scanning through the frequencies of excitation. In the future, it is intended to address the effectiveness of a multimodal excitation to damp a combustion instability by offering, at once, several frequency bands where the acoustic energy can be dissipated.

The chosen modes compose the first inversion of a major chord, like playing E-G-C on a piano. There are no multiples of each other, with their relative periods being 5, 6, and 8. This is the reason why a multimodal wheel having 25, 30, and 40 holes or teeth positioned at different radii was designed and tested, as shown in Figure 8.

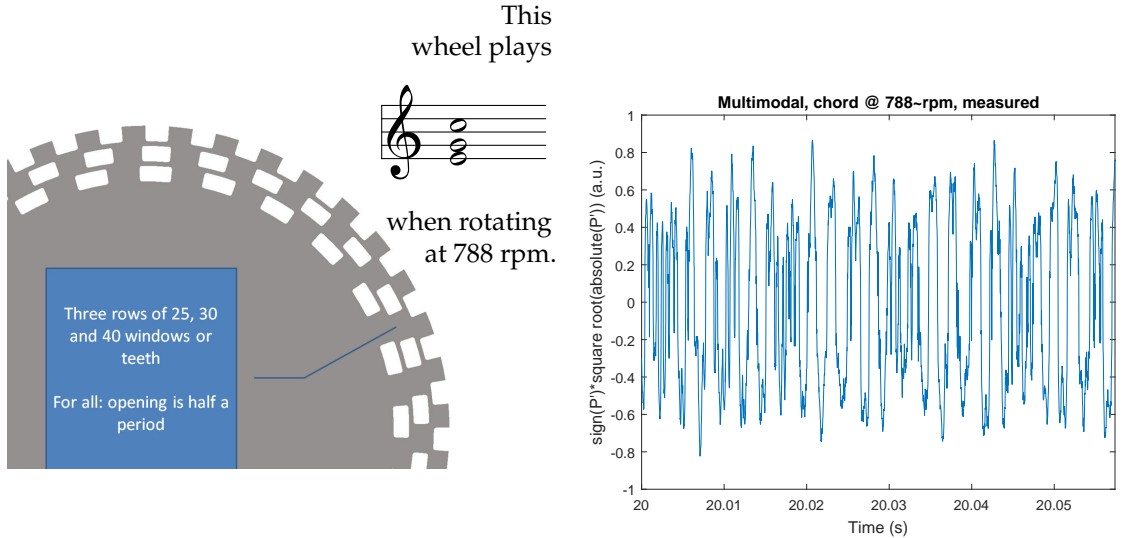

**Figure 8.** The multimodal wheel, designed to generate a synthetic signal reproducing a major chord, and measured chord waveform.

The device combines with the nozzle slot shown before. For this section, the reference diameter is 12 mm and the surface is one quarter of a round. The amplitude feature can be used to play none, one, two, or three tones at a time. The shown chunk from Figure 8 is derived from the measurement of the dynamic pressure 0.5 m down the siren's nozzle.

### 3.4. Nozzle Free Surface Calculation and Pulsed Flow Prediction Tools

In the following, the variable section $S$ is determined. The case of a round nozzle of critical diameter $d$ is shown.

The parameters are the eccenter, the diameter of the nozzle, and the lower and higher diameters of the cogged wheel. Based on the tilting angle of the chassis, the eccenter will vary, and the coverage of the nozzle will change accordingly. The whole kinematics are reconstructed, fed by the previous parameters. The eccenter is the distance separating the motor axis from the cogged wheel axis, it has a fixed value. It is set by rotating an off-centre chassis, as shown in Figure 9, where two cases are shown. The nozzle and the wheels are interchangeable as a function of the type of pulsation which is desired. The figure shows how, by completely retracting the wheel, and how, by penetrating the jet to its maximum, the sonic jet is alternatively opened (no tooth blocking the path) and closed (tooth blocking the path). The first wheel is such that it does only partially cover the nozzle with a maximum pulsation of say 50%; it is called the safety wheel and is used in explorative cases, where the resonance frequencies of the system are not known. Once these are identified and can be avoided by passing through at zero gain, the full range-wheel can be used, which covers up to 100% of the nozzle's critical surface during the pulsation cycle.

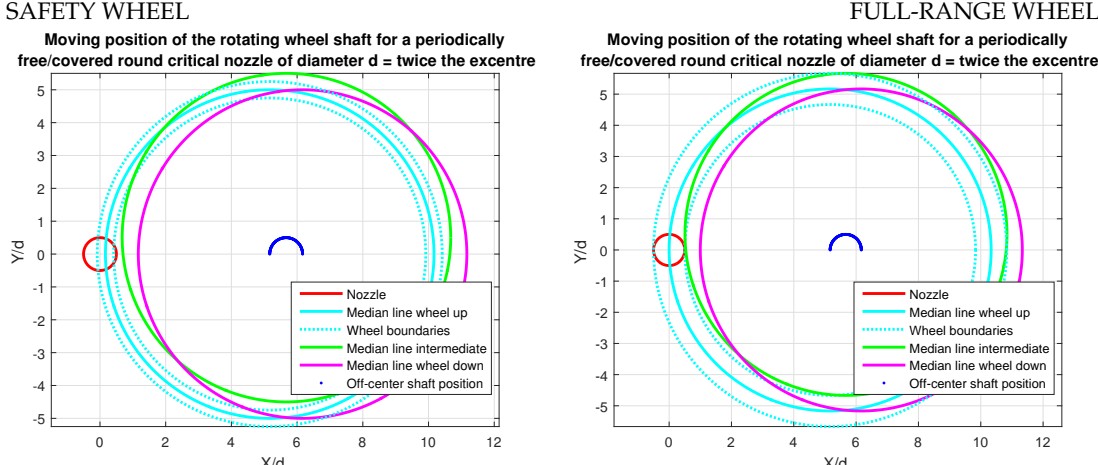

**Figure 9.** Relative position of the cogged wheel (mobile) versus the nozzle surface (fixed), for the safety and full-range versions.

In the case where the diameter $d$ is twice larger than the eccenter, the pulsation levels and resulting average mass flow can be derived as shown in Figure 10. A pulsation amplitude by $x$% means that, during the pulsation cycle, $x$% of the critical surface $S$ is covered. Assuming constant $(P_0, T_0)$ generating conditions, the resulting mass flow fluctuation is shown.

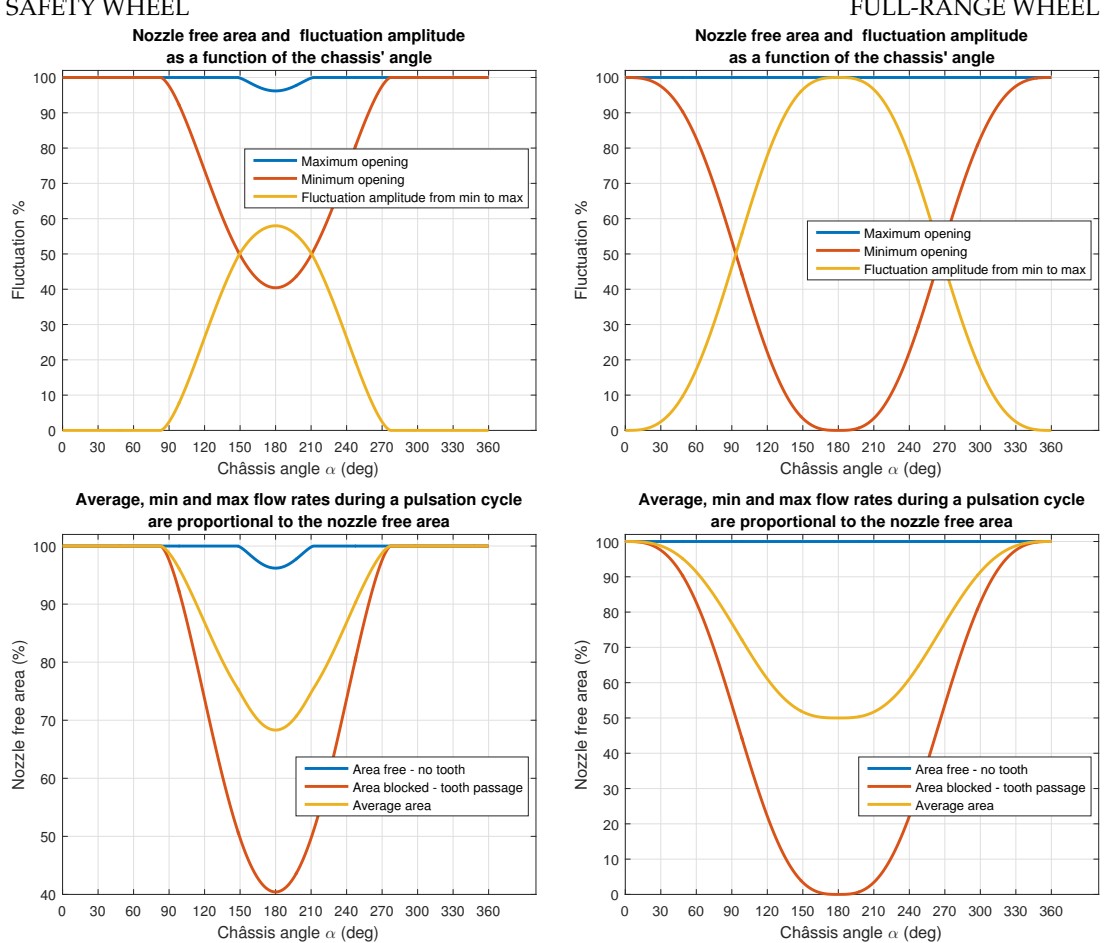

**Figure 10.** Percentage of covered surface of the jet critical section expressed as a function of the tilting angle $\alpha$ of the off-centre frame, and corresponding mass flows during the pulsation cycle.

The coverage of the nozzle's surface is calculated numerically as a function of the nozzle and wheel geometries, in order to reconstitute the mass flow pulsation pattern at the siren's outlet, and is based on the previous feedback of the amplitude. Early works were reported by A. Lang [35], who tested, among others, cogged wheels with a sine profile within his PhD Thesis.

The results of the prediction tool are shown in Figure 11. Round nozzles periodically covered by step-like teeth will generate sine-like patterns (as confirmed with Laser Doppler Anemometer (LDA) measurements in Reference [26]) when the pulsation level is low or when the tooth width is small compared to the nozzle diameter. When the coverage tends towards 100%, the signal tends towards a saw pattern when the tooth period is near to twice the width $w$ of the slot, and towards a square signal when the tooth period is much greater than $w$. A multimodal wheel was also computed reproducing a major chord, for good comparison with the last section of this article dedicated to this subject. The last signal chunk shown in Figure 11 compares well with the signal measurement from Figure 8, despite the fact that it is unfiltered (jet noise and 50 Hz disturbance are present). Nevertheless, the pseudo-periodicity, the peak count, and the component slides can be recognised.

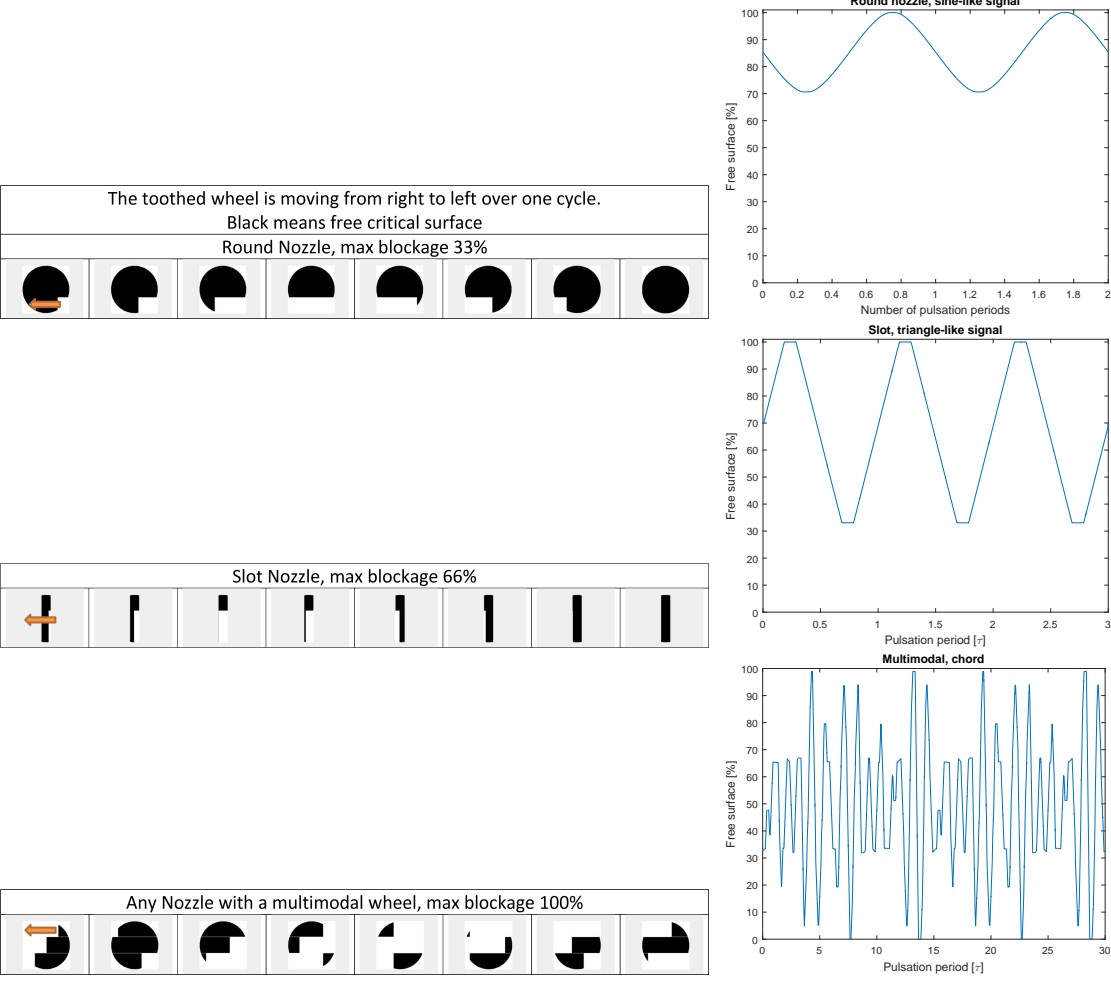

**Figure 11.** Module "Eclipse" for the calculation of the pulsation pattern. (**Top**): Sine-like signal with a round nozzle at 33% coverage. (**Middle**): Saw-like modulation with a quarter-round slot at 66% coverage. (**Bottom**): Multimodal wheel (major chord).

*3.5. Siren Performance*

The pulsation frequency is function of the number of teeth on the wheel and of the electric drive performance. The pulsation frequency relates to the shaft's rotational speed $\Omega$ (rpm) as follows:

$$f = \frac{n\Omega}{60}. \tag{4}$$

CBOne's siren 3G is equipped with a 250 W DC motor driving the cogged wheel (topping at 6000 rpm) and a geared 60 W step motor moving the chassis. The compact design responds to the demands of different laboratories, including hot flow conditions. The focus was put on the flexibility aspect, as well as on the robustness, allowing 4-h-long combustion tests sessions in a row. The device is air-cooled. The standard input/output flanges are DN 50/40 on the high pressure and low pressure side. The aggregate's dimensions L × W × H are 600 mm × 210 mm × 230 mm for a weight of 15 kg. The performance parameters are shown in Table 1.

**Table 1.** Performance sheet of the third generation siren. Values in bold are theoretical; others, including between parenthesis, were proven experimentally.

| Parameter | | Lower | Design | Full Load |
|---|---|---|---|---|
| Frequency range | (Hz) | 0–500 | 0–1000 | 0–6000 |
| Number of teeth $n$ | – | 5 | 10 | 60 |
| Pulsation | (%) | 0 | 30 | 100 |
| Nozzle critical surface | (mm$^2$) | 20 | 50 | 120 |
| Free flowing mass flow at room temperature | (g/s) of dry air | 10 | 50 | **500** (120) |
| Maximum admissible pulsation level | (%) | 100 | 100 | **5** |
| Corresponding SPL at outlet stagnation point under atmospheric conditions | (dB SPL) | 113 | 146 | **155** (150 forced resonance [36]) |
| Open/close Periodicity | (%/%) of cycle | **20/80** | 50/50 | **80/20** |
| High pressure side | (bar abs) | 2 | 4 | **20** (12) |
| Low pressure side | (bar abs) | 1 | 2 | **10** (6) |
| Flow temperature | (K) | **223 (dry)** | 293 | 550 |
| Accelerations/ decelerations | (Hz/s) | 1 | 30 | 60 |
| Amplitude change rate | (%/s) of full scale | 1 | 2 | 4 |
| Frequency uncertainty | (%) | ± 1 | ± 1 | ± 1 |
| Average amplitude uncertainty | (%) of full scale | ± 3 | ± 3 | ± 3 |

The uncertainty on the frequency depends on the precision of the electric drive only. In case a high-precision is requested in the low-frequency range, it is recommended to work with the least number of teeth. Phase-control using this device is feasible in the 0–200 Hz. The error term in terms of amplitude is specific of the maximum covered surface over the whole actuation range. Typical values are reported in the table.

## 4. Calibration of Dynamic Sensors and Acoustic Characterisation of the Hot Core

### 4.1. Calibration of Dynamic Sensors

The siren can be used for dynamic sensor calibration. A solution based on the siren technology would be an alternative to the usual piston calibrators, with an enlarged operation envelope. It can achieve, in a well-controlled manner, high-power pulsations and visit intermediate pressure and temperature conditions on the sensor's membrane. As shown in Figure 12, the sensor (fast-pressure transducer, or accelerometer) is placed at the tip of a total pressure probeholder facing the siren's jet. The principle of retractable wheel is kept, so that the set-up's resonant modes can be avoided. This set-up would be the first of its kind, opening the door to a new type of extreme conditions testing.

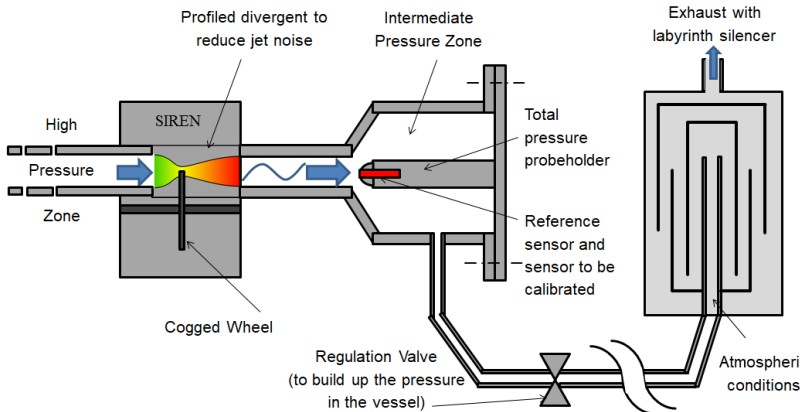

**Figure 12.** Schematic set-up for dynamic sensor calibration at various intermediate pressures and temperatures.

The set-up from Figure 12 was tested using the Vibro-Meter measurement chain detailed in reference [37] (MEGGITT, Fribourg, Switzerland). The same fast-pressure sensing equipment is used in the next sections. The results are shown in Figure 13. The nozzle used is a slot nozzle (reference diameter 10 mm, surface 1/4 of a round) to explore high frequencies, with a 20-teeth wheel so that the expected signal should be near to a triangle signal. The probe is situated in the jet of a 26 mm diameter pipe, and the probe itself has a diameter of 13 mm. The probe is at ambient conditions. The generating pressure $P_0$ is 2.8 bar. The pulsation is set to all-or-nothing (or 0 to 100% surface blockage during the cycle). The total pressure fluctuation at the probe's surface corresponds to the change in dynamic pressure (2.25 mbar amplitude, or 138 dB SPL).

By scanning through the frequencies, one detects peak amplitudes situated between 1.5 and 4 mbar. The explanation offered for the discrepancies is that the blockage of the probe in the jet is high, thus reducing the outlet cross section. Furthermore, some quarter-wave modes and subharmonics are met, which could explain the gain up to a factor of 2. On the other hand, some changes in cross section in the set-up are responsible for Helmholtz effects, and the absorption at some frequencies can be heard. Another explanation is the possible non-linearity of the sensor used, operating in its bottom-end region of scale, while the conversion from electrical current to pressure value was entered as a fixed factor.

Nevertheless, these results are very encouraging. The expected order of magnitude is met. This first test shows that the expected triangle-like signal is well reproduced. These results are well-repeatable within a 4% uncertainty range. The calibration is made possible up to 6 kHz in a next step, making this benchmarking method unique.

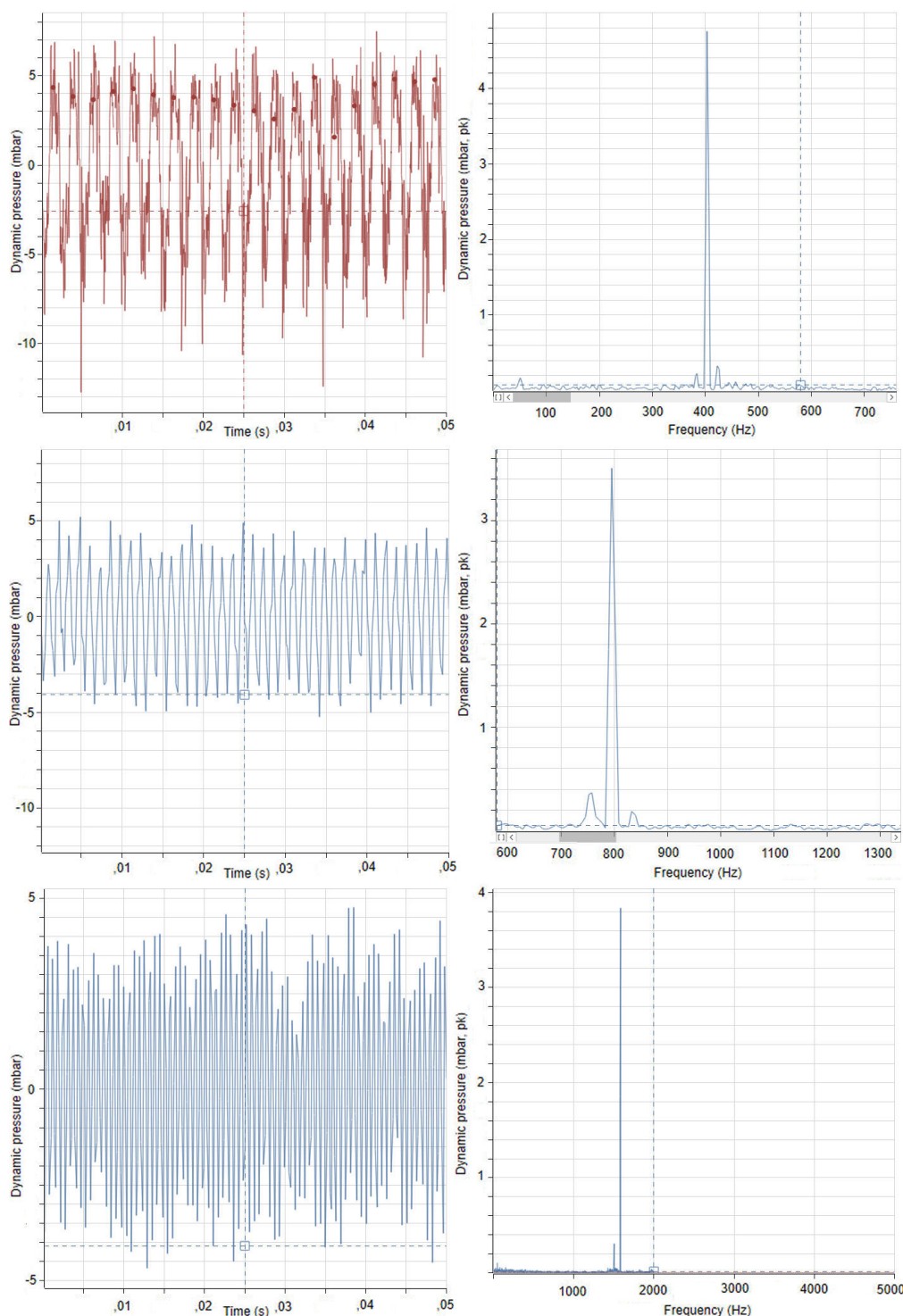

**Figure 13.** Highest amplitudes met on a 0–2 kHz calibration process. reported are the frequencies 402, 795, and 1680 Hz. (**Left**) waveform, (**Right**) amplitude spectrum.

### 4.2. Determination of the Eigenfrequencies and Related Time Lags in the Combustor

The flame transfer function, or the combustor acoustic response under cold flow conditions are determined experimentally using the siren as an exciter and a set of fast-pressure transducer distributed along the combustor (or resonator) measuring the core's response to the excitation. The vibro-meter chain used in this paper has up to 6 fast pressures transducers CP232 (high temperature resistant, flush-mounted on the pressure casing, sensitivity 800 pC/bar) and 2 accelerometers of type CA134

(outer skin-mounted, sensitivity 10 pC/g, see the specific results in Reference [37]). The acquisition device is a VM600 rack including a XMV16/XIO16T card pair. It can acquire signals with an acquisition frequency up to 100 kHz over 16 channels coded in 24 bit. The data monitoring and post-processing is done with the VibroSight software suite (all technologies by Vibro-Meter, MEGGITT, Fribourg, Switzerland).

The siren can "play" pre-programmed batch files containing a sequence of different excitation. This is done first under non-reactive conditions and then in the presence of a flame. The typical sequence is shown in Figure 14. The first part of the experiment contains frequency ramps performed at fixed amplitude (possibly several of them up to zero amplitude to differentiate sound from structure vibration). After that, the frequencies of interest that repeat well during the ramps are investigated one by one, with a variable amplitude in order to perform a sensitivity analysis and a determination of the gain. This experiment is called "Kölner Dom" in tribute to the Cathedral of Cologne that happens to match well the typical pattern of the scanned frequency.

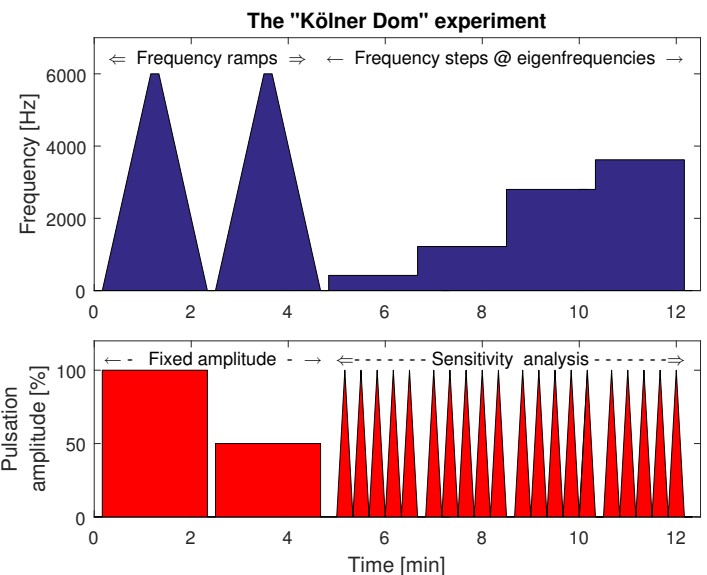

**Figure 14.** The "Kölner Dom" experiment, or the typical sequence of excitation of the siren for an acoustic characterisation of the system.

The method is applied to an experimental combustion test rig called "the emootion facility" [38] equipped with an additively manufactured burner made up of Inconel 718 called "CBO4" [39]. The combination of both, as well as the specific instrumentation, are presented in a separate paper [40]. This is a single-sector pressurised combustor including a two-staged, axially centred burner with a transparent liner allowing an optical access to the flame. The siren nozzle is a 12 mm quarter-round nozzle.

The response of three CP232 probes are shown. These are mounted at several locations on the pressure casing of the combustor, as suggested in Figure 17. The sensors are positioned so that the sensor called Dyn01 is diametrically opposed to Dyn03, flush mounted on the pressure casing at a distance of 20 mm downstream of the front plate. Dyn02 is in line with Dyn03 at a distance of 40 mm of the front plate.

The cold flow characterisation using the Kölner Dom method is shown in Figure 15. The upper plot shows the good reproducibility of the method by repeating three times in a row the same sequence. The resonant frequencies found during the ramps are 180, 606, 1440, and 1960 Hz. These are analysed after that as frequency steps in the frame of the sensitivity analysis.

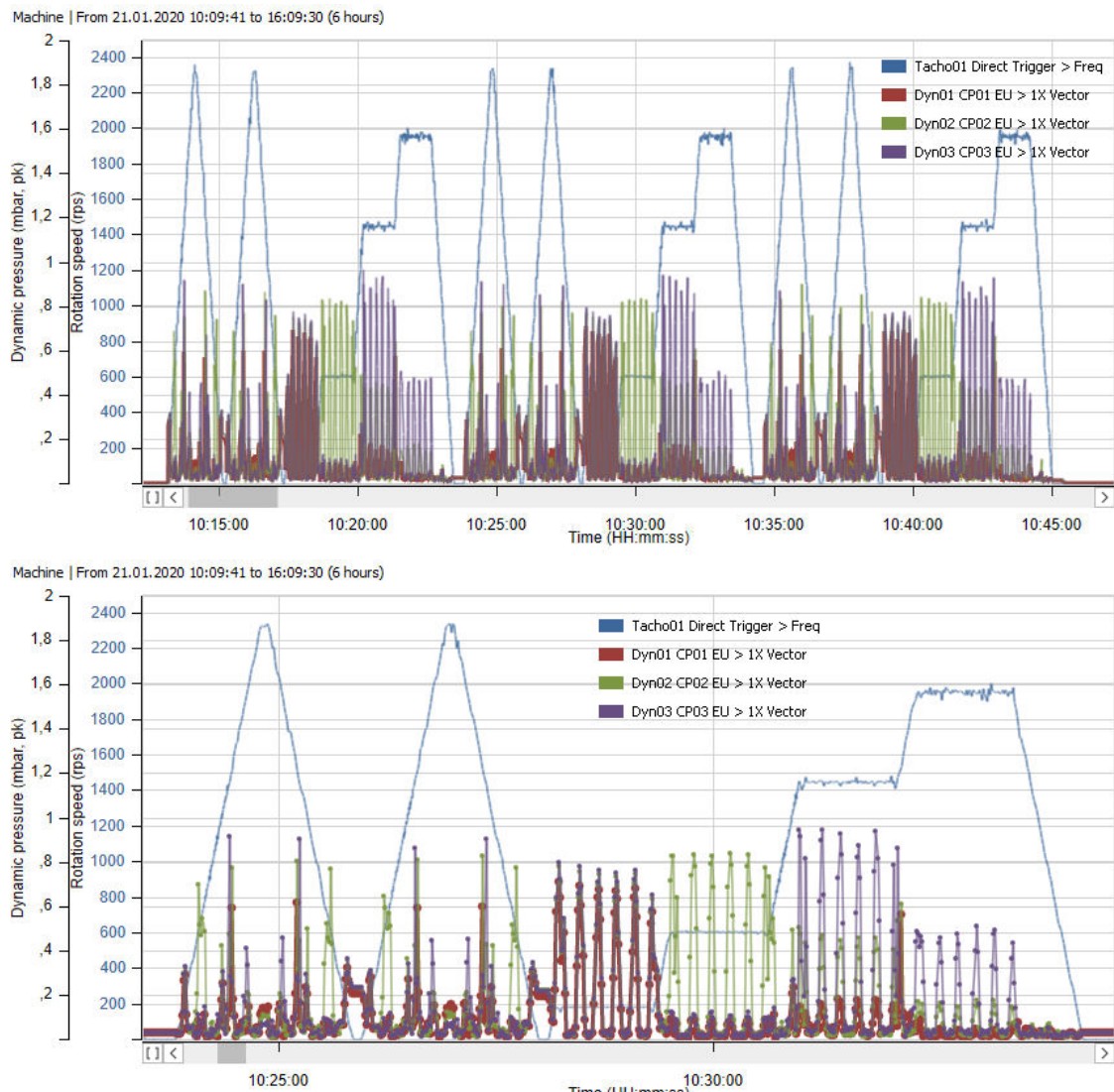

**Figure 15.** "Kölner Dom" measurement performed on the emootion test facility [38] with non-reactive tests at standard conditions, nozzle 12 mm. (**Top**): Repetition 3 times of the characterisation sequence to verify the reproducibility. (**Bottom**): Focus on one measurement.

What can be observed from these measurements are probably standing waves, for instance, at 180 Hz, where all sensors must be placed near a belly (same excitation pattern, same amplitude), while the different measurement locations become visible with the further frequencies.

### 4.3. Modal Analysis: Measurement of the Phase Shifts or Time Lags in Relationship to Standing or Transported Waves

The same experiment is repeated with a 6 kW premixed flame in the same facility (see Figure 16). One focuses on the 1604 Hz resonance, which is observed in presence of a flame, and gets amplified by the siren's excitation. The sensitivity analysis shows that the excitation in all-or-nothing mode brings the acoustic pressure from 0 at rest near 1 mbar peak. The phase shifts versus the synchronisation signal (TTL of the siren) are displayed, too: These can be measured with accuracy in the lower kHz domain based on a 10 kHz sampling frequency, and repeat well during the fluctuations of amplitude. Based on that, a multiple sensor analysis is possible in order to understand the type of instability (axial or tangential, standing or travelling wave).

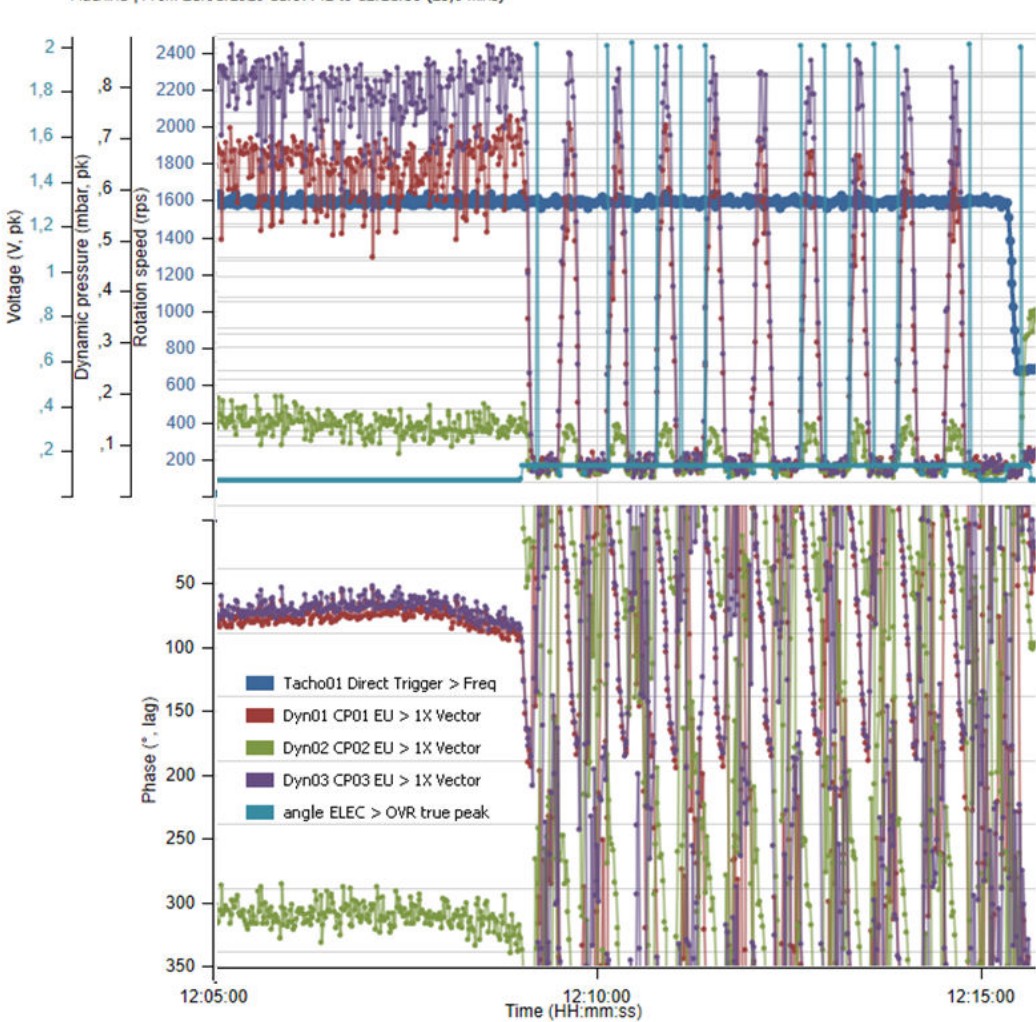

**Figure 16.** Excitation of the flame at 1604 Hz, where a resonance was observed with combustion. The first part is a continuous excitation; the second part is a sensitivity analysis with variable amplitude. The top plot reports the peak amplitudes of the fast pressure transducers. The lower plot reports the phase shift measured at this frequency versus the reference signal of the pulsator.

### 4.4. Health Diagnostic of Fast Pressure Sensors and Accelerometers during Operation

This is a field application for power gas turbines. It can be performed at rest or during operation and does not necessitate to dismount the sensors, which is convenient. The siren is mounted in blow-down modus (see Figure 17) and generates a calibrated noise at non-critical frequency when the machine is running, or scans through frequencies when the machine is at rest. A high-frequency band is suggested for this purpose in the same figure, where calibrated noise could be generated during operation, without a risk of triggering a combustion instability. Indeed, no significant resonance was observed in the 2.4 kHz+ zone while exciting with the siren up to 6 kHz. This is a way to test how the dynamic sensors (fast pressure transducers, accelerometers) respond, without having to dismantle them, and monitor their response over time within an acceptable drift margin.

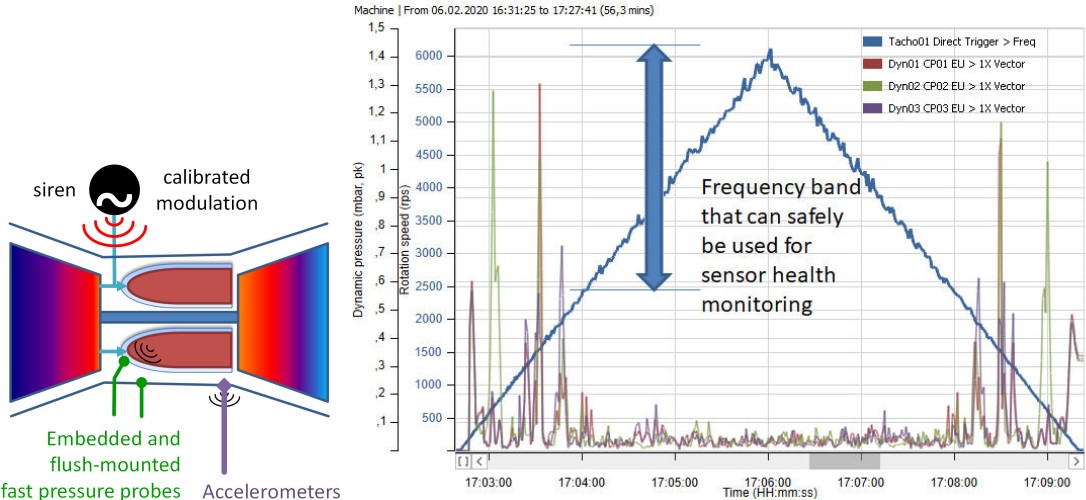

**Figure 17.** Health diagnostic of the GT's dynamic sensors during operation. (**Left**): Arrangement. (**Right**): Suggested hi-frequency excitation over the 0-6k Hz range, outside of the bands where acoustic couplings are observed.

### 4.5. Modal Analysis Using the Multimodal Excitation

Figure 18 shows the power and phase spectra in a run with three consecutive ramps up and down in frequency using the multimodal wheel from Figure 8. The fast pressure transducer is situated 1/2 m down the siren's nozzle. A masque hiding the sound power level situated below twice the average of the whole take was applied to filter out these plots. The three frequencies and their phase information are well visible and well repeatable.

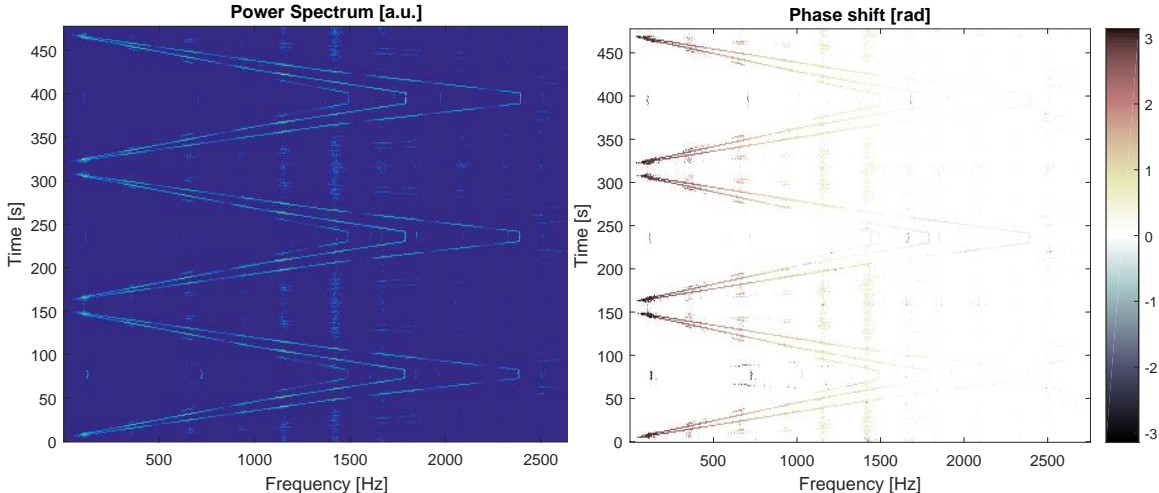

**Figure 18.** Three consecutive frequency scans (ramp up and down) using the multimodal wheel. (**Left**): Power spectrum, (**right**): Phase information.

The phase information is of importance, because it is related to the source of noise. In Figure 18, the sole source of noise is the siren itself. To highlight this effect, a Helmholtz resonator was placed near the fast pressure probe, under the same experimental conditions (Figure 19). The dimensions of the resonator are the following. The bulk volume $V$ is 300 mL, the outlet orifice $A$ has a round diameter of 18 mm, and the length of the neck is 80 mm. The speed of sound $c$ is 340 m/s at room conditions. The resonant frequency $f_H$ is estimated as follows:

$$f_H = \frac{c}{2\pi}\sqrt{\frac{A}{VL}}. \tag{5}$$

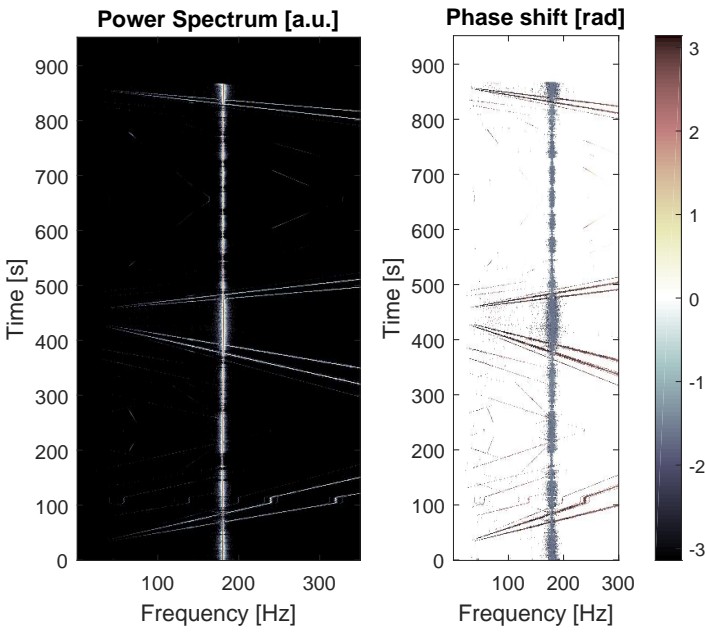

**Figure 19.** Scanning through the bulk mode of an Helmholtz resonator.

The substitution into the previous formula produces an estimated resonant frequency $f_H = 176$ Hz. The measured resonance takes place in bulk mode centred at 182 Hz with a variance of 4 Hz, matching well with the previous estimate. The effect of the siren, when each of the excitation frequencies meets the Helmholtz' own, is to narrow the bulk effect and amplify the peak frequency. However, the Helmholtz cavity remains as the main source of noise at 182 Hz as shown by the phase trace.

The advantage of using the multimodal wheel for scans is therefore that the information on the resonant frequencies, and a hint on the resonators location is multiplied by the numbers of modes on the wheel for one take. Furthermore, provided the highest mode is phase-locked with the siren (here, the outer crown of 40 teeth), the phase difference between all modes is constant as a function of the frequency, as long as the siren is the noise source. This means that an observed shift in phase versus a set value corresponds to the excitation of another part in the machinery (e.g., the flame) that overrides the sound generated by the siren. Instead of applying the multi-microphone method to situate a sound source [41,42], only one sensor monitors single sources emitting several known frequencies (the siren, and any resonant part excited by the siren). The microphone can be phase-locked on the pattern period, meaning every five counts of the lower frequency wheel, six counts of the middle, or height counts of the outer one (effectively: this one). Therefore, in theory, one sensor suffices to monitor the machine, and redundancy is no longer required. In practice, the number of dynamic sensors can be reduced to the least number meeting the safety and certification requests.

## 5. Combustor Flow Control and Flame Forcing

### 5.1. Effective Flow Control in the Combustor of a Gas Turbine: Possible Siren Placements

In all previous works by the same team, the siren was used in so-called blow-down configuration, where at least a part of the combustion air is being pulsed upstream from the burner. Regarding real gas turbines, having an additional piece of equipment that needs a separate pressurised air feed with backpressure at least double that of the largest operating pressure ratio (OPR) might not be wanted. A more convenient way to use the siren is the so-called discharge mode, where a part of the plenum air is sampled periodically and derived a bleed towards the siren. By comparison to the blow-down excitation, the discharge provokes only a periodic change in mass flow, but theoretically no sound, because sound does not travel upstream in a transonic configuration. Thus, it is a pure

aerodynamic fluctuation. The column of air flowing through the vessel is brought to resonance, and these resonances—or eigenfrequencies—are specific of the component and/or of the local conditions exposed to this flow modulation. The air bleed can be used for other purposes, such as service air or cooling air. Both possible uses of the siren actuator in blow-down and discharge configuration are shown in Figure 20.

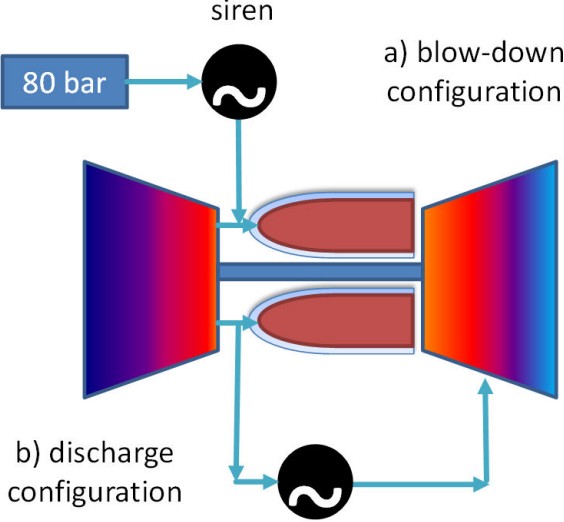

**Figure 20.** Possible implementations of a siren on a gas turbine for flow control [36]: (**a**) Blow-down and (**b**) discharge configurations.

The tests presented under this section were performed with the discharge configuration, which gives a less strong disturbance to the combustion chamber volume. The idea is to decouple the noise emitted by the siren from the noise emitted by any resonant element in the pressure vessel. In the meantime, the periodic bleed is used as an excitation to reveal and enhance these resonances. In the end, the objective is to excite the flame effectively, without overriding it because of the pulsation's exaggerated energy level. Figure 21 shows the experimental arrangement.

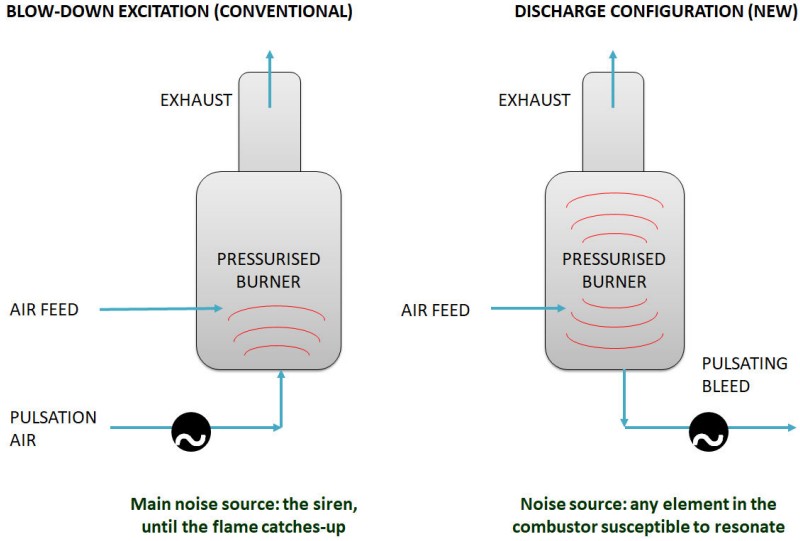

**Figure 21.** Functionality of siren configurations on the pressurised test rig.

The proof of concept of the discharge strategy is shown in Figure 22. The MethaNull test rig is a single-sector combustion set-up for confined flames with optical access, for single- or multi-stage burners. It was adapted for intermediate pressure measurements with a 5 mm outlet nozzle. The burner model CBO4 is a compact two-stage premixed module produced with additive manufacturing (selective laser melting of Inconel 718 powder), operated with propane mixture at 7 kW under 2.3 bar abs. Details about the test rig and burner can be found in References [39,43]. The siren releases into the ambient a bleed performed in the plenum. Sound and light intensity signals from the flame were recorded by two separate devices, so that no cross-talk is possible. The details on the optical technique used to corroborate the fast pressure measurement can be found in a separate paper by same authors [40]. The on-off excitation appears clearly on the sound signature and more faintly on the photodiode sensor, so that the flame resonates effectively at 525 Hz.

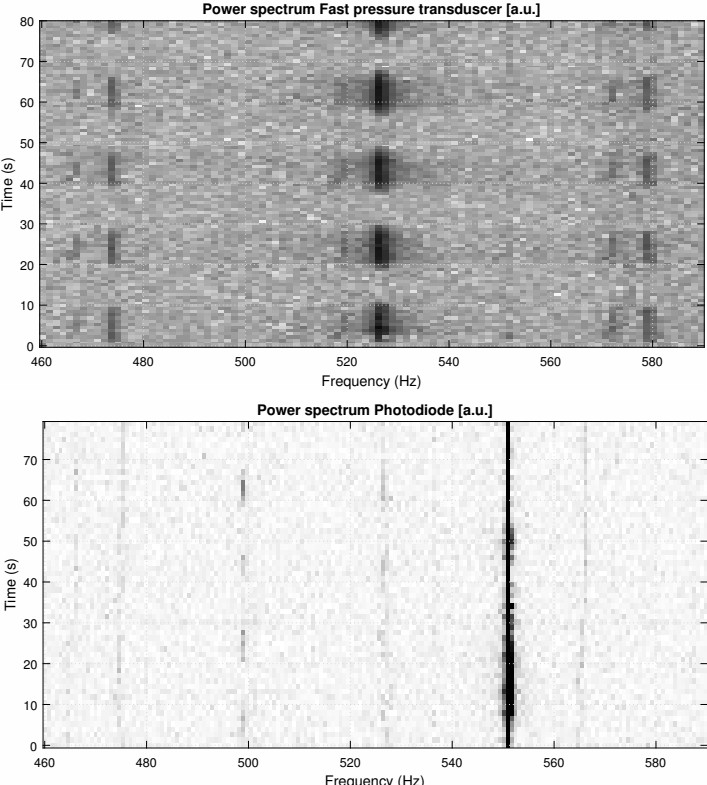

**Figure 22.** Effective on-off excitation of the confined flame at 525 Hz at 2.3 bar. (**Top**) Readings of the fast pressure sensor. (**Bottom**) Reading of the optical sensor of the RCP probe [44].

The current research focuses on a more effective drive using the discharge effect. The challenge is to sample the smallest possible quantity of bleed air as near as possible to the burner, without risking to aspire fuel.

### 5.2. Analysis of a Pulse Jet

The siren has been used to describe the multiphase flow dynamics in the primary zone of a single-sector combustor equipped with a laboratory airblast-burner in the presence of combustion instabilities [19,27].

Figure 23 shows the air flow dynamics, measured with multicomponent LDA at the non-reactive operation point. 3D-Streamlines are computed in the gathered flow-field starting from where the fuel should be injected. The pulsation frequency is 100 Hz. The cyclic pattern shows a change in the angle of the injection that will result as a back-and-forth motion of the internal recirculation zone; meanwhile, a ring vortex on the outer detaches from the front plate and is transported by the flow.

Figure 24 highlights the dynamics of the ring vortices (or "donuts") being generated around the jet at the front plate, detaching the front plate, and then being transported by the flow. The coherent structures in the flow are revealed using the second invariant criterion (called also $\lambda_2$ criterion) from Jeong and Hussain [45]. Figure 24 uses the same "data brick" of LDA measurements as required for computing the streamlines from Figure 23 and focuses on the extreme jet velocities to materialise the jet envelope and the different recirculating zones, the vorticity (based on a curl of the velocity field), or the axial gradient. This is a way to materialise the ring vortex in 3D.

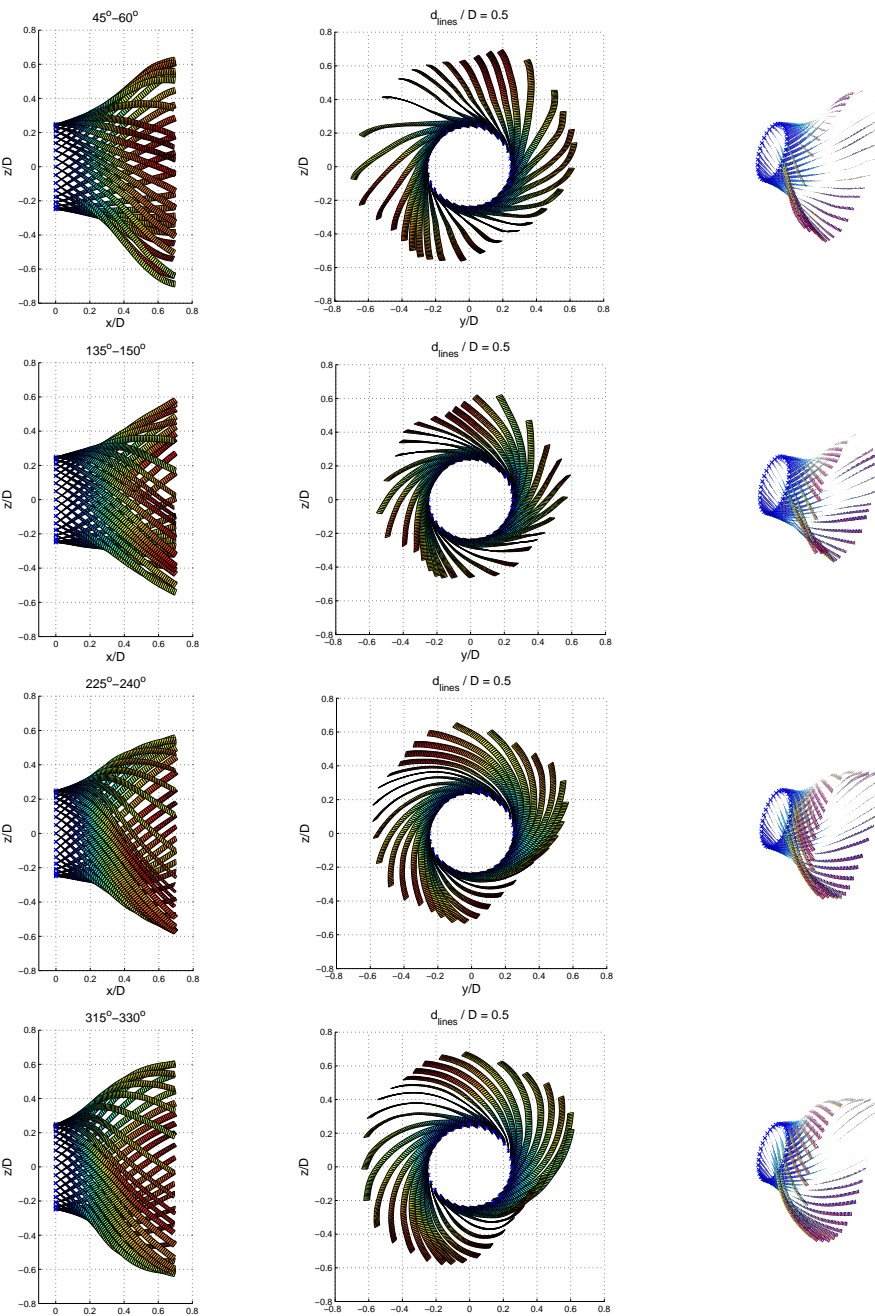

**Figure 23.** 3D streamlines for free-jet configuration, emitted along a $d/D = 1/2$ diameter circle placed in the injection section (mixing layer, where the liquid phase is injected). (**Left** and **middle**): Profile and front views of the 3D streamlines. (**Right**): three-quarter view. Air pulsation 30% of 30 g/s at 100 Hz.

About 24 h of LDA measurements where needed to gather all the measurement points needed to generate the displays in Figures 23 and 24. The siren allowed the flow pattern to be kept well-repeatable and provided the synchronisation signal to reproduce in a refined way the whole pulsation cycle.

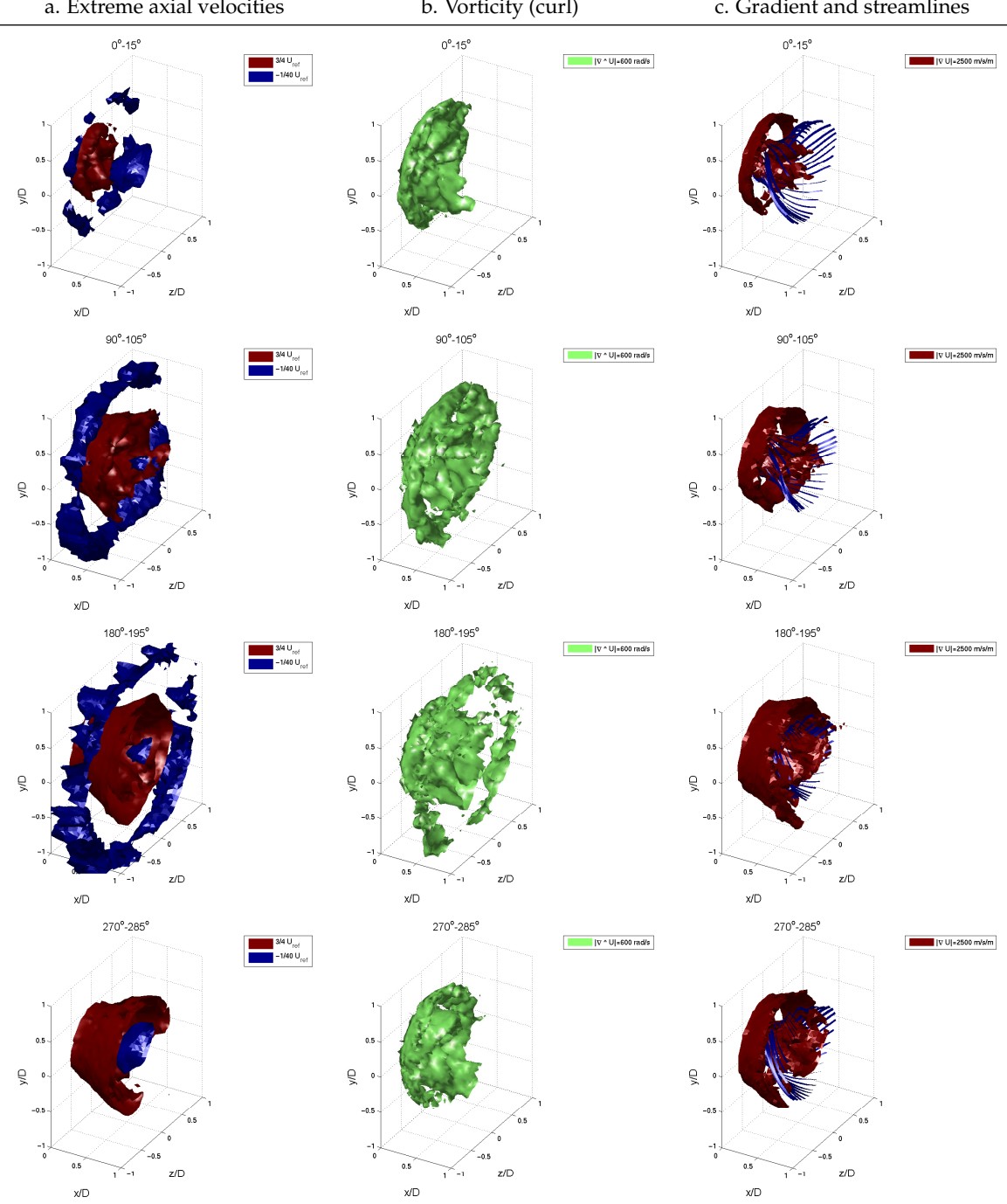

**Figure 24.** 3D analysis of the airblast aerodynamics, measured with conditioned Laser Doppler Anemometer (LDA) and phase-averaged [19,27]. (**a**) Extreme axial velocities *u* to sort out the air jet shape and the recirculating zones. (**b**) Analysis of the field vorticity, applying the curl operator $\nabla \wedge \mathbf{U(u,v,w)}$. (**c**) Analysis of the mixing layer positions (gradient operator and normalisation: $|\nabla \mathbf{U(u,v,w)}|$) comparison with the 3D streamlines emitted from the liquid injection

### 5.3. Modal Control of a Pulse Flame

The siren has been used to excite the resonant modes of a premixed air-methane burner, built as quarter-wave resonator (Figure 25) [28,30]. With a 3 m long pipe separating the siren's nozzle from the burner, the fundamental harmonic is 25 Hz, and only the subharmonics of frequency $f = 25 \times (1 + N)$, where $N = \{1, 2, 3, ...\}$, can be amplified.

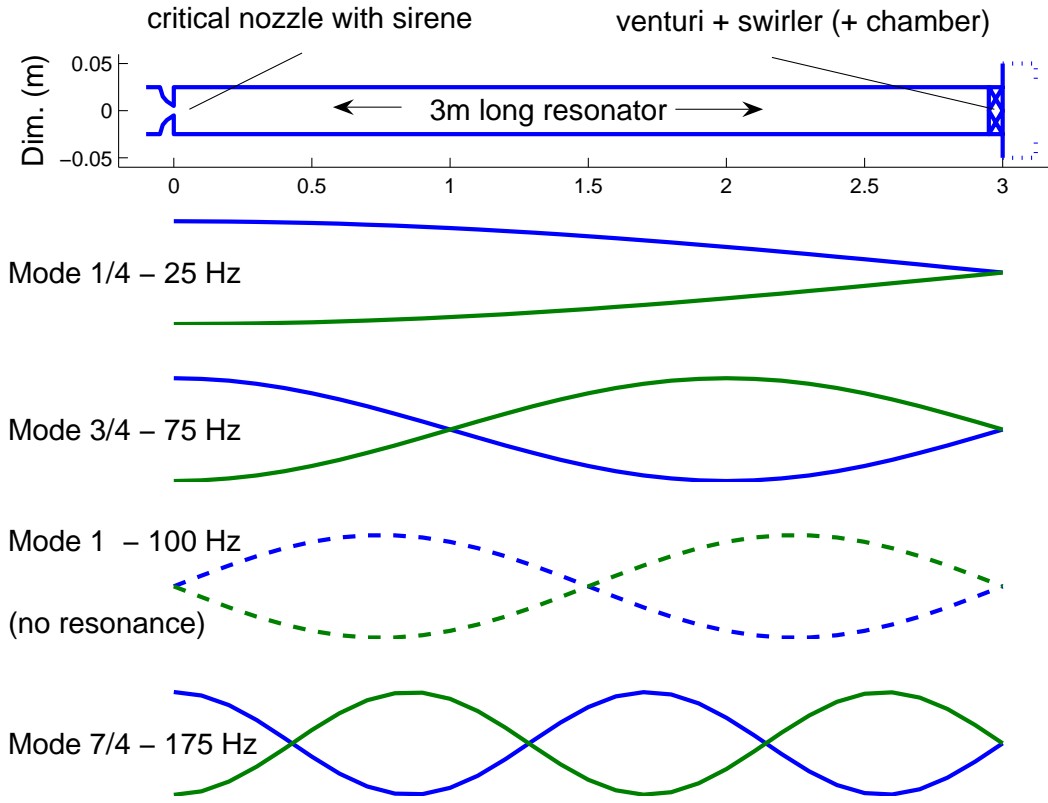

**Figure 25.** Amplified modes with the siren on a quarter-wave resonator.

The flame would receive an actuation at any frequency as shown in the previous section, but the one meeting one of these resonant frequencies are remarkable because the flame catches-up and becomes the main source of noise. The pulsation dynamic can be observed in Figure 26. The first column described with a schlieren technique the V-shaped swirl flame, attached to the tip of the burner. The middle column shows the 25 Hz resonance. The outer contour of the flame creates a recurrent roll driven by the detachment of a peripheral ring vortex. Although the burners are not the same, the fresh reactants jet dynamics and the outer vortex ring detachment are similar, respectively, to those of Figures 23 and 24, a detailed particle image velocimetry (PIV) analysis published in Reference [30] establishes this fact. The deformation of the 25 Hz flame is impressive. This is definitely the type of instability that, when self-sustained, would harm a combustor. The siren can be used to force a subharmonic, like the frequency 175 Hz as shown with the right column. At this pulsation frequency the flame remains bounded where the steady state flame used to be. The deformations observed have a much lower amplitude than observed in the 25 Hz pulsating case. The flame envelope remains bounded within its original domain. This suggests the idea of modal control that can be driven by the siren: Move the dynamic of a flame to a higher frequency and reduce the wavelengths so that, even though resonant, the flame does not harm the structure.

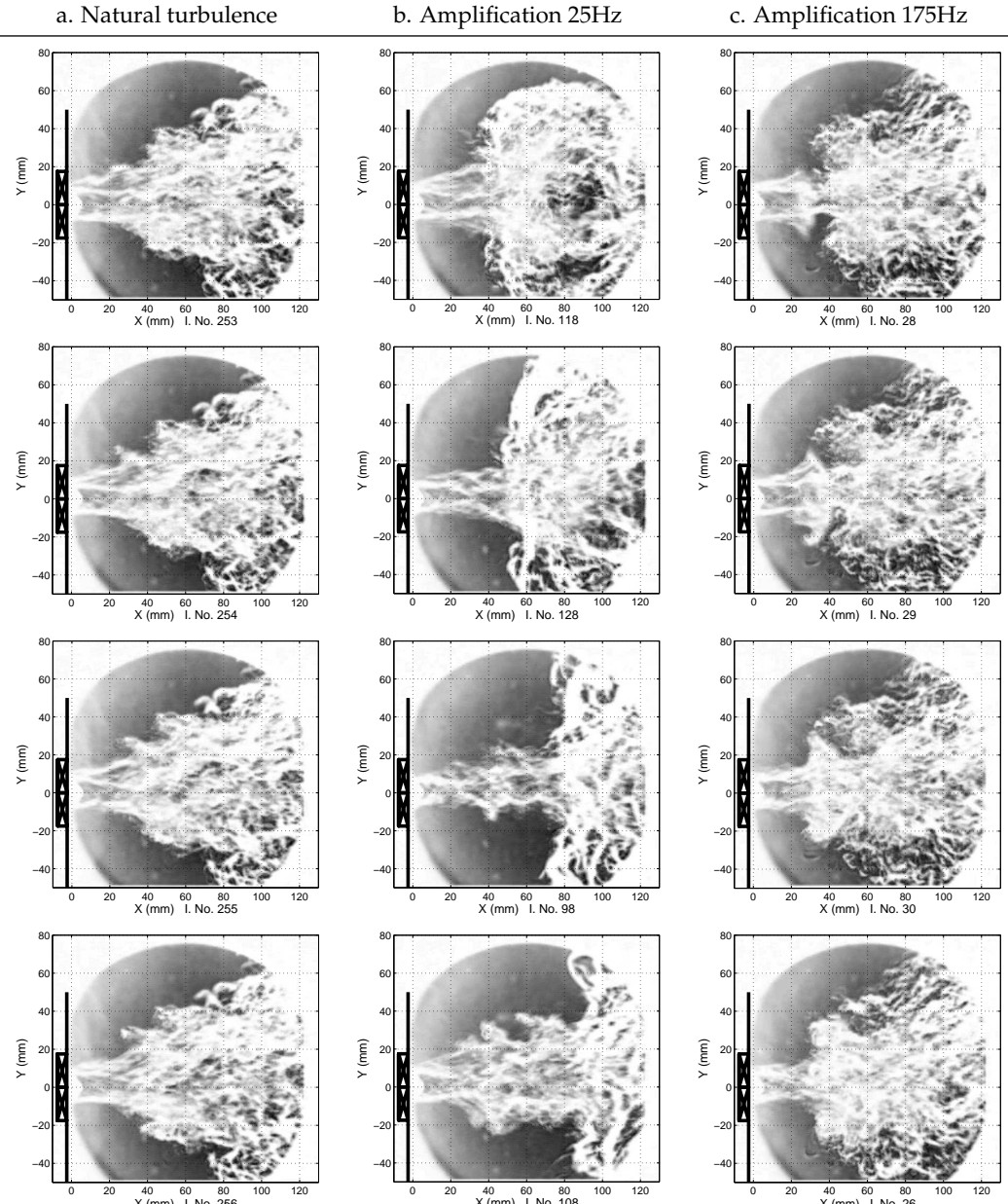

**Figure 26.** High speed schlieren visualisations (1000 fps, single frame, full sensor resolution) on a premixed air-methane flame in free-jet configuration. Column (**a**): Turbulent flame. (**b**): Amplification 25 Hz. (**c**): Amplification 175 Hz. All consecutive pictures are chosen and arranged so that one complete pulsation cycle is visible for the modes 25 and 175 Hz. Inverted grayscale.

### 5.4. Improvement of the Combustion Performance Using Forcing

It was observed from the previous study that the 175 Hz pulsed flame offered a better robustness versus lean blow-out limit than the naturally turbulent flame (without forcing) [43]. The offered explanation is that the deformation of the pulse flame is such that the flame front concentrates on a smaller volume and augments the density of energy, so that forcing at a rightly chosen frequency can contribute to a better self-sustain of the flame. This feature was explored and confirmed in the study of Reference [36], where an operation point at part load known to operate poorly was improved with help of the siren.

Figure 27 shows the pre-programmed "Kölner Dom" sequence of the testing procedure using the siren, repeated twice to highlight the similarity in the response of the flame. At first, a ramp in

frequency is performed to detect the eigenfrequencies of the flame (also see Reference [43] for the scanning and peak-finding procedure). After that, each resonant frequency is assessed (200, 368, and 512 Hz) with variable amplitudes on the configuration of the cited paper. This is the sensibility analysis, in order to determine the amount of energy to be brought in the pulsation to drive the flame. At 368 Hz pulsation, both NOx and CO emissions could be reduced by 15%, while the combustion efficiency impaired at 98% at steady state could be brought back to 100%.

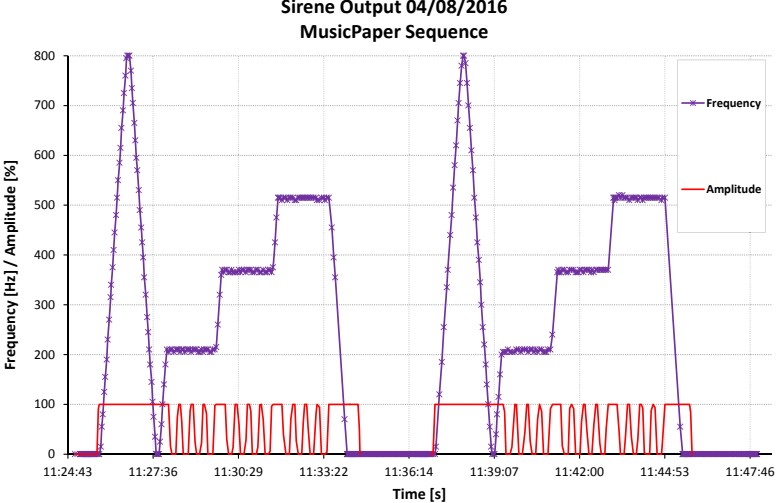

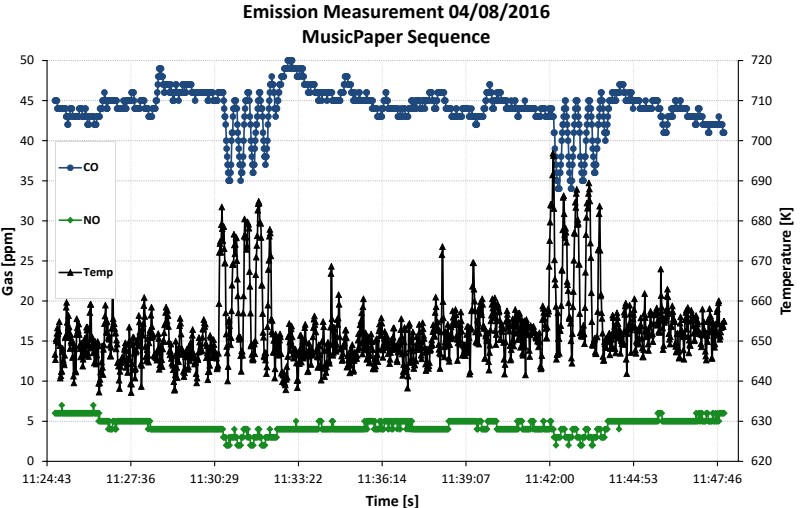

**Figure 27.** Test sequence using MUSICPAPER control software. (**Top**) Time-sequence repeated twice, starting with 2 frequency ramps followed by 3 frequency steps, at variable amplitudes of pulsation. (**Bottom**) Simultaneous exhaust gas measurements.

## 6. Future Developments: A More Powerful Siren

Solutions are currently being tested to augment the pulsation performance while keeping the pulsator's dimensions similar. Over the last years, requests about higher frequencies and higher pulsation levels have been formulated. The higher frequency is needed to visit up to the second harmonic of precessing vortex cores [18,32] or spinning tangential instabilities [33]. The second is to

achieve pulsation levels near 155 dB SPL for calibration purposes, and at higher ambient pressures and temperatures than standard conditions.

In the short term, we intend to produce the Siren-E ("E" for energy) covering this type of applications. Among the modifications, a profiled jet divergent should reduce the jet noise. The pressure performance will be quadrupled: the high pressure port will go up to 80 bar, the low up to 40 bar, and the pressure step will be dimensioned for up to 40 bar. The device is intended to be compatible with many existing gas turbines.

The siren-E is also envisioned as a sub-part of a dynamic sensor calibration set-up. A solution based on the siren technology would be an alternative to the usual piston calibrators, with an enlarged operation envelope. It can achieve in a well-controlled manner high-power pulsations and visit intermediate pressure and temperature conditions on the sensor's membrane. The multimodal wheel allows the monitoring of several peaks, which are not harmonics of each other, generated by the siren. The principle of retractable wheel is kept, so that the set-up's resonant modes can be avoided. This set-up would be a first of its kind, opening the door to a new type of extreme conditions testing.

## 7. Conclusions

The article summarises about 20 years of research on calibrated flow forcing using three successive generations of siren, including the latest developments on high-frequency and multimodal excitation. The background of this study is on combustion stability. The different options for flow forcing, the specifics, the dimensioning details, and the successive features of a siren were described in detail. Examples of use show how effective low-energy forcing can be on injection dynamics. Some ideas were proposed, such as the passivation of strong combustion instabilities using modal control by displacing the resonance energy to a higher harmonic, more diffusive and less harmful, or acoustic energy dissipation using the multimodal excitation. It was also shown that the discharge configuration is feasible as equipment on a power gas turbine and effective. The siren generation will be an adaption of the current to higher energy levels, compliant with power and laboratory applications. It will therefore be more powerful as a new calibration method and as a realistic and reliable flow controller for safe and quiet gas turbine operation.

## 8. Patents

Giuliani, F. Vorrichtung und Verfahren zum Betreiben einer Flamme (apparatus and method for operating a flame), 2016. Patent AT516424B1.

**Author Contributions:** F.G.: Conceptualization, methodology, supervision, writing. M.S.: hardware and software aspects. N.P.: data acquisition and processing. L.A.: experimental research and laboratory curation. All authors have read and agreed to the published version of the manuscript.

**Funding:** The progress in the siren technology is supported financially since 2016 by the FFG (Austrian Research Promotion Agency) and by the BMVIT (Austrian Federal Ministry for Transport, Innovation and Technology) in the frame of the collaborative research "emo(o)tion" (Engine health monitoring and refined combustion control based on optical diagnostic techniques embedded in the combustor, contract 861004, consortium Combustion Bay one e.U. and FH JOANNEUM) in the frame of the "Take-Off" program.

**Acknowledgments:** Excerpts of works performed within the DGA/PEA TITAN (Projet d'Etude Approfondie de la Délégation Générale de l'Armement, France, 1999–2002), within the EU-funded Project NEWAC (New Aeroengine Core Concepts, FP6 2006–2010, contract AIP5-CT-2006-030876), and within the project MethaNull (financially supported by the JITU-PreSeed Grant of the Austrian Federal Ministry for Economy, Family and Youth under guidance of the AustriaWirtschaftsservice GmbH (AWS) from 2013 to 2015, Contract P1302031-PSI01). Special thanks to Andreas Lang, Pierre Gajan and Krzysztof Solinski for their kind help. We hereby wish to acknowledge all parties.

**Conflicts of Interest:** The authors declare no conflict of interest.

## Abbreviations

The following abbreviations are used in this manuscript:

| | | |
|---|---|---|
| CBOne | | Combustion Bay One e.U. |
| CBO4 | | Name of the burner described in Reference [39] |
| $d$ | (mm) | Nozzle diameter (round) or reference diameter (slot) |
| DN | (mm) | Nominal pipe size (from *diamètre nominal*) according to standard ISO 6708 |
| emootion | | Project acronym, stands for "engine health monitoring and refined combustion control based on optical diagnostic techniques embedded in the combustor" |
| $f$ | (Hz) | Frequency |
| $f_H$ | (Hz) | Resonant frequency in an Helmholtz resonator |
| LDA | | Laser Doppler Anemometer |
| $\dot{m}$ | (kg/s) | Air mass flow |
| $n$ | | A given number, number of teeth |
| $r$ | (J/kg/K) | Air constant, $r = 286$ J/kg/K |
| OPR | | Operating pressure ratio |
| PIV | | Particle image velocimetry |
| $P_0, T_0$ | (Pa,K) | Generating pressure and temperature upstream of the chocked nozzle |
| $S$ | (mm$^2$) | Area of the critical section |
| USP | | Unique selling proposition |
| $x$ | | Variable |
| $w$ | (m) | Width of the slot |
| $\alpha$ | (rad) | Angle deciding the opening of the nozzle's slot |
| $\gamma$ | (-) | Heat capacity ratio of air |
| $\omega$ | (rpm) | Spool rotating speed |

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
