# Peer review of "Forcing Pulsations by Means of a Siren for Gas Turbine Applications"

_ijtpp, doi:10.3390/ijtpp5020009_

Round 1
Reviewer 1 Report
The article presents the research activity conducted over the last 20 years by the Graz University group on calibrated flow forcing using three successive generations of siren, including the latest developments on high-frequency and multimodal excitation.
In general, the work presents a clear structure and is interesting. It presents the working principles of the siren, shows the first versions of sirens from which the research team started for the next upgrades and improvements.
Probably the title can be misleading and the reviewer suggests to change it. From the original title, in fact, the reader expects a review of the siren system in general, while the article covers a few applications and focuses on the particular case, although of interest, of the work done by the authors or resulting from it.
Reading a review, the reader may be interested in having an idea of applications of gas turbine sirens, in finding a review of siren design solutions, in getting an idea of the theory behind it and an explanation of the principle of operation. As this is not a general discussion, these aspects are missing or confined to the particular case under examination.
Or you may be interested in getting an idea of complications, problems of use and perhaps solutions to them if you have to make an assessment of the adoption of such a system or have problems of operability. This last section is missing and it would be interesting to add it in this work as well.
Just to mention a few aspects: non-linarites that may occur during the excitement. Noise at frequencies other than that of excitement. If you place the shut valve in a rather narrow tube after the siren, we can get a wave steepening phenomenon if the amplitude is sufficiently high and the tube sufficiently long...
The reviewer suggests the article for journal publication with some minor review to be done following the comments provided. Specific comments are also reported in the section below.
Specific Comments:
Introduction:
- Line 23. “Steady" means unstructured, although turbulent. Not clear the meaning of this steatment. Please, clarify.
- Line 28. algebraic sum of Vector sum?
- Initially the authors talk about the Joule-Bryton cycle and machines... so the reader understand the turbine in general. In the following explanation the author focuses on combustion instabilities, assuming to restrict the field to the combustor. The passage is not clear and needs improvement, and the explanation given, although correct, is not supported by a precise scientific technical language. It´s recommended to improve. Just a couple of general examples: "the flame acquires a great dynamic", "The sum of it all degrades the performance of the machine" (it leads to risks of integrity and / or requires a reduction of the operating window or load ...).
2.2 Active control
- The principle is clear, but two comments are necessary. The first is that it does not emerge the fact that in practice in the application power there are practically never solutions of this kind. The second is missing an example where this solution is successfully used: description of the problem and how the active control introduces the solution.
3.2 Technical solution to achieve elevated frequencies
- Lines 2015-2017 The maximum achievable frequency… not clear why the maximum frequency is related to these parameters. Intuitively one would say that the frequency, for a pure harmonic wave, would depend on the maximum rotational speed of the wheel and the tooth period.
- Figure 8 - show fft to see the frequencies in the signal and the relative amplitude. How to make sure that they do not interact non-linearly?
- This is an important point and the authors should dedicate a few words to clarify it. Comment on how increasing the power of excitement can achieve non-linearity. What are the identified limits, how are they managed? How is it controlled? The reviewer imagines that this is an even more serious problem in the case of multimodal excitation.
- Figure 11- As per Figure 8, please, show the fft of the signal
3.5. Siren performance
- Formula 4 - not very clear what this pulsation frequency actually represents. Is this the period of the periodic wave generated? If so, in case of generation of a sine-like signal it also coincides with the frequency in the signal itself. In case of other signals, such as the saw-tooth signal or multi-harmonic one, what does this frequency represent? or better to say, how does this relate with the frequency content in the signal?
4.1. Calibration of dynamic sensors
- Line 303 The generating pressure is 2.8 bar. where is this pressure taken? indicate it in the scheme shown
- Lines 314-317 This kind of conclusions or presentation is not what one expects in a review. It is more paper-like. I think it´s good to keep them but just to stress the fact again that the title should be adapted to better capture the nature of the article.
4.2. Determination of the eigenfrequencies in the combustor
- Line 329 Fig 16. error in reference? Please, show a picture of the investigated combustor indicating also the sensor placement and the boundary conditions realized at the outlet. This would help the reader´s understanding also in the following part
- Line 331- not clear from the picture the resonant frequencies found
4.3. Modal analysis: measurement of the phase shifts or time lags in relationship to standing or transported waves
- Line 342 1604 Hz resonance which is observed in presence of a flame not clear... is this is a self-excited dynamic?
- Line 435 measured with accuracy in the kHz domain what about lower frequencies? Please, add a statement.
4.5. Modal analysis using the multimodal excitation
- Line 367 -368 Helmholtz cavity remains as the main source of noise Not clear to the reviewer: when the siren frequency meets the combustor frequency then we should see higher amplitude in the spectrogram. But when the experiment is repeated with the Helmholtz damper tuned at that frequency, then the amplitude should be lower because of its damping effect.
- It´s not clear why the damper should act as a source and enforce amplitudes. Why high amplitudes are observed at the damper frequency (fig 19) at each instant in time, even when the siren is not exciting that frequency?
- Figure 15 - improve quality to better see all the information pointed in the text
- Figure 18: missing scale for amplitude. Correct Up-bottom with left right. The fact that straight lines are seen for the resonant frequencies at each instant in time means that the siren is continuously exciting them. Please comment on the impact on measurement quality.
- Line 410 GT2020-16007 Put in reference section
5.3. Modal control of a pulse flame
- Lines 440-442 This suggests the idea of modal control that can be driven by the siren: move the dynamic of a flame to a higher frequency and reduce the wavelengths so that, even though resonant, the flame does not harm the structure
Not clear to the reviewer what the authors mean. How are the dynamics moved to higher frequencies? Just exciting at a higher (resonant) frequency?
As I interpreted the statement, I would disagree. In fact, independently on the siren excitation, if a disturbance is generated somewhere and start a feedback loop with the flame, an instability occurs and the limit cycle is reached. If in this case the siren excites a higher frequency this will happen on top of the previous process and will not prevent the instability to occur. The process, until the limit cycle is reached can be, in fact, considered linear. What could be done with a siren is to excite at the same frequency of the instability, out of phase so that the driving force is cancelled. But this would be "active control".
5.4. Improvement of the combustion performance using forcing
- Line 444 It was observed in the previous study that the 175 Hz pulsed flame offered a better robustness versus lean blow-out limit than the naturally turbulent flame (without forcing). Where is this shown?
- Line 451 Please correct answer with response
- Line 452 detect the eigenfrequencies of the flame please add explanation on how the eigenfrequencies of the flame are defined and how are they retrieved with the measurement technique
- Line 453 visited What does it mean?
- Line 455 turn the flame unstable. when is the flame defined unstable? please report definition
- More details, results and explanation need to be added in this section
Author Response
Dear Mrs Zhuang!
Dear Reviewers!
First of all, thank you for your excellent reviews and comments on our works, which, I am sure, will help us to produce an interesting article.
Second, I apologyse for the delay in this rebuttal. It is just that in Europe everything is incredibly slow at the moment. My company is still running with people working at home, and I spend my life repeating the same thing 12 times a day in a different manner over different communication tools... I hope you went through this coronacrisis without damage, and whish you good health!
The rebuttal follows.
With kind regards,
For the authors
Fabrice Giuliani
---------------------
GLOBAL COMMENTS
- This paper is instrumental to a separate paper submitted at the same time and currently under revision
Manuscript ID: ijtpp-758879
Title: Combined optic-acoustic monitoring of combustion in a gas turbine
Authors: Fabrice Giuliani *, Lukas Andracher, Vanessa Moosbrugger, Nina Paulitsch, Andrea Hofer
Received: 13 March 2020
This revised paper better highlights the connexion to the separate paper. - The English was improved
- The description of the results was improved as requested
- One common comment to all reviews is the misleading term "Review" in the title. Actually the article starts as a review and then compiles the results of one specific technology. We therefore decided to clarify the title so that
(previous) "A review on forcing pulsations by means of a siren for gas turbine applications"
becomes
(new) "Forcing pulsations by means of a siren for gas turbine applications" - Furthermore the contribution of Andreas Lang (cited in the aknowledgment in the first version) in this rebuttal is more than substantial. Fairly, Mr Lang was moved in the author list. I hope that this is ok with you?
--------------- Point by Point Reviewer 1
Introduction:
- Line 23. “Steady" means unstructured, although turbulent. Not clear the meaning of this steatment. Please, clarify.
It is a common mistake I meet during my lecture, that students mix the notions of steady state and laminar flow. I felt compelled to insist on the fact that a steady flow can be turbulent. The sentence was removed.
- Line 28. algebraic sum of Vector sum?
Yes indeed. The algebraic sum is for 1D flow in pipes. Thanks for that.
- Initially the authors talk about the Joule-Bryton cycle and machines... so the reader understand the turbine in general. In the following explanation the author focuses on combustion instabilities, assuming to restrict the field to the combustor. The passage is not clear and needs improvement, and the explanation given, although correct, is not supported by a precise scientific technical language. It´s recommended to improve. Just a couple of general examples: "the flame acquires a great dynamic", "The sum of it all degrades the performance of the machine" (it leads to risks of integrity and / or requires a reduction of the operating window or load ...).
The paragraph was reformulated so that the GT technology is the background and that solving the problem of combustion instabilities is the incentive.
2.2 Active control
- The principle is clear, but two comments are necessary. The first is that it does not emerge the fact that in practice in the application power there are practically never solutions of this kind. The second is missing an example where this solution is successfully used: description of the problem and how the active control introduces the solution.
We comment prior in the text that passive control techniques (damping) are always preferred to active - that augment the complexity and costs.
One well-known application is cited : "The first successful industrial combustion stability control systems were acting on the injection of fuel, such as Moog’s D6xx series Direct-Drive Valves (DDV) used by Hermann et al. [3]."
Actually, thermoacoutics are more visible on the heat generation side (useage, not avoidance). This paragraph was rewritten, including more examples about pulse combustion.
3.2 Technical solution to achieve elevated frequencies
- Lines 2015-2017 The maximum achievable frequency… not clear why the maximum frequency is related to these parameters. Intuitively one would say that the frequency, for a pure harmonic wave, would depend on the maximum rotational speed of the wheel and the tooth period.
You are right. This regards the maximum number of teeth on a wheel, in order to achieve high frequencies. The limiting factor is the ratio between the tooth period and the nozzle width. The correct notion is the cut-off frequency. As soon as a tooth period is smaller than the width of a jet, the cut-off frequency is met. The paragraph was rewritten accordingly.
- Figure 8 - show fft to see the frequencies in the signal and the relative amplitude. How to make sure that they do not interact non-linearly?
The FFT is visible on figure 18 under the shape of a spectrogram. The multimodal wheel ramps up and down three times and shows three distinct excitation traces. The phase-shift shows distinct time lags, so that there is a priori no interaction. By taking period ratios of 5/8, 6/8 and 8/8, one expects to avoid interactions. These would be more likely to appear if sub-harmonics were excited, such as 2/8, 4/8 and 8/8.
- This is an important point and the authors should dedicate a few words to clarify it. Comment on how increasing the power of excitement can achieve non-linearity. What are the identified limits, how are they managed? How is it controlled? The reviewer imagines that this is an even more serious problem in the case of multimodal excitation.
This question is very general, while the minimum excitation amplitude needed to achieve an non-linear growth is system specific. Physically, the energy addition pro cycle must be higher than the energy dissipation (or damping). This also why it is convenient to have a system where the amplitude of excitation can be modulated.
The idea of the multimodal wheel is essentially to accelerate the scanning process (figure 18). Indeed it can be argued, how peaks interact with each others in the low frequency range. But while moving in the upper frequency domain (a few kHz) the peaks are distinct.
- Figure 11- As per Figure 8, please, show the fft of the signal
See figure 13 (single peak) and 18 (multimodal)
3.5. Siren performance
- Formula 4 - not very clear what this pulsation frequency actually represents. Is this the period of the periodic wave generated? If so, in case of generation of a sine-like signal it also coincides with the frequency in the signal itself. In case of other signals, such as the saw-tooth signal or multi-harmonic one, what does this frequency represent? or better to say, how does this relate with the frequency content in the signal?
You mentionned it earlier: this is the frequency of pulsation, as a function of the nember of teeth on the wheel, and the wheel's rotating speed.
4.1. Calibration of dynamic sensors
- Line 303 The generating pressure is 2.8 bar. where is this pressure taken? indicate it in the scheme shown
That would be the pressure of the air on the high pressure side of figures 4-5-6 and extensively discussed in chapter 3. We added "The generating pressure Po measured on the high-pressure side of the siren " to clarify.
- Lines 314-317 This kind of conclusions or presentation is not what one expects in a review. It is more paper-like. I think it´s good to keep them but just to stress the fact again that the title should be adapted to better capture the nature of the article.
It is more like progress than review, this is true. See our preamble.
4.2. Determination of the eigenfrequencies in the combustor
- Line 329 Fig 16. error in reference? Please, show a picture of the investigated combustor indicating also the sensor placement and the boundary conditions realized at the outlet. This would help the reader´s understanding also in the following part
Pardon, the mentionned frequencies are specific to Fig. 15 (error of cross-reference, corrected).
Since we discuss a general method we preferred not to show a specific rig geometry. The dynamic sensors are flush-mounted on the combustor casing as shown in figure 17. The test rig in question is extensively described in the separate paper IJTPP mentionned in the preamble. The connexion is highlighted in the revised paragraph, with citation.
- Line 331- not clear from the picture the resonant frequencies found
Pardon, the mentionned frequencies are specific to Fig. 15 (error of cross-reference, corrected). Fig 16 details the sensitivity analysis performed for the frequency 1604Hz under combustion conditions. The paragraphs were corrected accordingly.
4.3. Modal analysis: measurement of the phase shifts or time lags in relationship to standing or transported waves
- Line 342 1604 Hz resonance which is observed in presence of a flame not clear... is this is a self-excited dynamic?
No it is not self-sustained. It gets excited by the siren when the frequency happen to match. This is observable during the frequency scans, and then investigated one by one for the relevant peaks. 1604Hz is one of them.
- Line 435 measured with accuracy in the kHz domain what about lower frequencies? Please, add a statement.
The comment is related to a measurement done at 1.6kHz. Lower frequencies are easier to investigate. The periods are longer and therefore they can be described with a greater accuracy (or greater amount of samples per cycle) than at higher frequencies (the least beeing 2 points per cycle, as dictated by the Shannon cut-of frequency) at a fixed sampling frequency. Here, operating at 10000Hz one has 2 points at 5000, 4 points at 2500 and 8 points a cycle at 1250 Hz. We were surprised to observe that these 4 to 8 points a cycle are "good enough" to repeat the time lags measured on the phase.
4.5. Modal analysis using the multimodal excitation
- Line 367 -368 Helmholtz cavity remains as the main source of noise Not clear to the reviewer: when the siren frequency meets the combustor frequency then we should see higher amplitude in the spectrogram. But when the experiment is repeated with the Helmholtz damper tuned at that frequency, then the amplitude should be lower because of its damping effect.
No - in that case the Helmholtz resonance is amplified.
What you mention is an Helmholtz resonator tuned to respond in bulk mode, over a bounded bandwidth. An accute resonance at a fixed frequency meeting this bandwidth will be damped because the energy is spread over a larger frequency band and brings a larger volume of fluid into motion, diluting the excitation energy in the first place and killing the accute resonance. That would be a different geometry.
- It´s not clear why the damper should act as a source and enforce amplitudes. Why high amplitudes are observed at the damper frequency (fig 19) at each instant in time, even when the siren is not exciting that frequency?
Because it is a resonator. It resonates all the time, as if blowing the tip of a glas bottle.
- Figure 15 - improve quality to better see all the information pointed in the text
Done
- Figure 18: missing scale for amplitude. Correct Up-bottom with left right. The fact that straight lines are seen for the resonant frequencies at each instant in time means that the siren is continuously exciting them. Please comment on the impact on measurement quality.
This is a spectrogram, the amplitude is arbitrary unit, and the intention is to show a binary information, where the trace of the siren is clearly visible at a given time for a given frequency.
The question on the measurement quality is important when the siren operates in transient mode (frequency scan or ramps). The rate of change in frequency must be set so that the scan is performed as fast as possible, but also so that the scan allows the system to resonate when being met. We usually operate the siren from 10 to 30 Hz/s rates on the ramps.
- Line 410 GT2020-16007 Put in reference section
done
5.3. Modal control of a pulse flame
- Lines 440-442 This suggests the idea of modal control that can be driven by the siren: move the dynamic of a flame to a higher frequency and reduce the wavelengths so that, even though resonant, the flame does not harm the structure
Not clear to the reviewer what the authors mean. How are the dynamics moved to higher frequencies? Just exciting at a higher (resonant) frequency?
Yes indeed. Here the "nasty" frequency is 25 Hz, the flame makes the noise of a chopper and accelerates the fatigue of the liner with time. By forcing a higher harmonic (175 Hz) the flame is still resonnant but its boundaries correspond more or less to the steady state flame. The thermal stress exerted on the liner is less harmull than in the first situation. Forcing the flame to move from the mode 1 to the mode 7 was positive.
As I interpreted the statement, I would disagree. In fact, independently on the siren excitation, if a disturbance is generated somewhere and start a feedback loop with the flame, an instability occurs and the limit cycle is reached. If in this case the siren excites a higher frequency this will happen on top of the previous process and will not prevent the instability to occur. The process, until the limit cycle is reached can be, in fact, considered linear. What could be done with a siren is to excite at the same frequency of the instability, out of phase so that the driving force is cancelled. But this would be "active control".
We both agree on the fact that a thermoacoustic coupling is taking place. Our point is that higher hamonic pulsations are "better" for the system than the fundamental modes that are self-pulsating. By acting so, the amplitude of motion of the flame is reduced. By moving towards higher frequencies, the effort in dissipation on the Kolmogorov scales is reduced as well. All in all, it's not great, but it is much better.
Let us assume that a critical pulsation takes place at some conditions, e.g. at cold start while speeding up on a large power gas turbine. This actuation can help moving through with less damage than conventionnally.
5.4. Improvement of the combustion performance using forcing
- Line 444 It was observed in the previous study that the 175 Hz pulsed flame offered a better robustness versus lean blow-out limit than the naturally turbulent flame (without forcing). Where is this shown?
Pardon. It was observed FROM the previous study... and the related reference with the 2015 and 2017 papers.
- Line 451 Please correct answer with response
Done, thanks
- Line 452 detect the eigenfrequencies of the flame please add explanation on how the eigenfrequencies of the flame are defined and how are they retrieved with the measurement technique
This is exactly the Kölner Dom method from Figure 14.
- Line 453 visited What does it mean?
assessed. Corrected
- Line 455 turn the flame unstable. when is the flame defined unstable? please report definition
... to drive the flame. This calls back to the first questions of yours: how great must be the excitation frequency so that the driving effect catches up. Corrected in the text.
- More details, results and explanation need to be added in this section
One figure about the jet dynamics was added, and the text was adapted accordingly. Thank you.
Submission Date 27 February 2020 Date of this review 21 Apr 2020 10:59:48

Reviewer 2 Report
In general sound and complete but for a review very focused on own work respectively siren model (title indicates a general review). For a review the siren and work from [22] for instance could be discussed in more detail and compared. In the present version the review part ends at 2.3, afterwards the focus is on a review of the past work. Abstract and title do not align well (e.g. line 13), description in line 56 aligns with title.
Flaws in structure:
- 2.4 -> new section 3 with title related to siren
- section 3 "elements of design" should be subsection of section 3
- section 4 should be subsection of new section related to applications
Other comments, please clarify or add minor parts:
- line 8: which pulsator?
- line 230: which study?
- line 292 and 471: are piston calibrators widely used? Systems with valves, pressure ramps and reference sensor, siren-like devices and shock tubes are used as well.
- line 299: which section about flame transfer function?
- line 322: remove "what"
- line 327: CBO4 burner ??
Author Response
Dear Reviewers!
First of all, thank you for your excellent reviews and comments on our works, which, I am sure, will help us to produce an interesting article.
Second, I apologyse for the delay in this rebuttal. It is just that in Europe everything is incredibly slow at the moment. My company is still running with people working at home, and I spend my life repeating the same thing 12 times a day in a different manner over different communication tools... I hope you went through this coronacrisis without damage, and whish you good health!
The rebuttal follows.
With kind regards,
For the authors
Fabrice Giuliani
---------------------
GLOBAL COMMENTS
- This paper is instrumental to a separate paper submitted at the same time and currently under revision
Manuscript ID: ijtpp-758879
Title: Combined optic-acoustic monitoring of combustion in a gas turbine
Authors: Fabrice Giuliani *, Lukas Andracher, Vanessa Moosbrugger, Nina Paulitsch, Andrea Hofer
Received: 13 March 2020
This revised paper better highlights the connexion to the separate paper. - The English was improved
- The description of the results was improved as requested
- One common comment to all reviews is the misleading term "Review" in the title. Actually the article starts as a review and then compiles the results of one specific technology. We therefore decided to clarify the title so that
(previous) "A review on forcing pulsations by means of a siren for gas turbine applications"
becomes
(new) "Forcing pulsations by means of a siren for gas turbine applications" - Furthermore the contribution of Andreas Lang (cited in the aknowledgment in the first version) in this rebuttal is more than substantial. Fairly, Mr Lang was moved in the author list. I hope that this is ok with you?
-----------------
In general sound and complete but for a review very focused on own work respectively siren model (title indicates a general review). For a review the siren and work from [22] for instance could be discussed in more detail and compared. In the present version the review part ends at 2.3, afterwards the focus is on a review of the past work.
Fixed. See the preamble.
Abstract and title do not align well (e.g. line 13), description in line 56 aligns with title.
Fixed. See the preamble.
Flaws in structure:
- 2.4 -> new section 3 with title related to siren
The idea was more to have chapter 2 speaking about all sorts of actuators. It ends with the one we further developed, the siren. Please note that the siren models mentioned do not all have a choked nozzle.
- section 3 "elements of design" should be subsection of section 3
From there on we focus on the siren with a choked nozzle, and its parameters.
- section 4 should be subsection of new section related to applications
Keeping the previous structure as such, we believe that a separate chapter on applications makes sense. Furthermore, results are shown in this chapter while previous is mostly state-of-the-art and theory. So that the clean cut is justified.
Other comments, please clarify or add minor parts:
- line 8: which pulsator?
Pulsator is a synonym of siren, and used to avoid repetitions. To make it clear, "... siren technology. The pulsator..." is replaced by "... siren technology. This pulsator..."
- line 230: which study?
"In this study..." is replaced by "In this paper..."
- line 292 and 471: are piston calibrators widely used? Systems with valves, pressure ramps and reference sensor, siren-like devices and shock tubes are used as well.
Yes indeed, but to the author's knowledge, the mostly deployed device is the piston calibrator, especially on field.
Can please reviewer 2 mention its sources where siren-like devices are used? We would be happy to complete our bibliography.
- line 299: which section about flame transfer function?
Thank you for spotting this one. The measurement chain mentioned is a MEGGITT vibrometer technology. The text is adapted accordingly.
The section "4.2 Determination of the eigenfrequencies in the combustor"
was renamed "4.2 Experimental determination of the eigenfrequencies and related time lags in the combustor" and starts with the sentence
"The flame transfer function, or the combustor acoustic response under cold flow are determined experimentally using the siren and a set of fast-pressure transducer. The vibro-meter chain used in this paper has up to 6 fast pressures transduscers CP232 (high temperature resistant, flush-mounted on the pressure casing, sensitivity 800 pC/bar) and 2 accelerometers of type CA134 (outer skin-mounted, sensitivity 10 pC/g, see the accelerometers results in [GT2016]). The acquisition device is a VM600 rack including a XMV16/XIO16T card pair. It can acquire signalsup with an acquisition frequency up to 100~kHz over 16 channels coded in 24 bit. The live- and post-processing are done with the VibroSight software suite (Vibro-Meter acquisition chain, MEGGITT, Fribourg, Switzerland)."
- line 322: remove "what"
Done
- line 327: CBO4 burner ??
Now described and referenced in the text. CBO4 is our monolotithic AM burner. It is premixed, two-stage, instrumented so that it determines in real time the air and fuel flow over pressure loss measurements, and the temperature as well. The ignitor is embedded. It is made-up of Inconel 718 and necessecitates very little finishig after production (socle separation, thermal treatment, and that's about it).
Thank you!
